# A Comprehensive Linear Speedup Analysis for Asynchronous Stochastic Parallel Optimization from Zeroth-Order to First-Order

**Xiangru Lian[*], Huan Zhang[†], Cho-Jui Hsieh[‡], Yijun Huang[*], and Ji Liu[*]**
[*] Department of Computer Science, University of Rochester, USA
[†] Department of Electrical and Computer Engineering, University of California, Davis, USA
[‡] Department of Computer Science, University of California, Davis, USA
xiangru@yandex.com, victzhang@gmail.com, chohsieh@ucdavis.edu,
huangyj0@gmail.com, ji.liu.uwisc@gmail.com

## Abstract

Asynchronous parallel optimization received substantial successes and extensive attention recently. One of core theoretical questions is how much speedup (or benefit) the asynchronous parallelization can bring to us. This paper provides a comprehensive and generic analysis to study the speedup property for a broad range of asynchronous parallel stochastic algorithms from the zeroth order to the first order methods. Our result recovers or improves existing analysis on special cases, provides more insights for understanding the asynchronous parallel behaviors, and suggests a novel asynchronous parallel zeroth order method for the first time. Our experiments provide novel applications of the proposed asynchronous parallel zeroth order method on hyper parameter tuning and model blending problems.

## 1 Introduction

Asynchronous parallel optimization received substantial successes and extensive attention recently, for example, [5, 25, 31, 33, 34, 37]. It has been used to solve various machine learning problems, such as deep learning [4, 7, 26, 36], matrix completion [25, 28, 34], SVM [15], linear systems [3, 21], PCA [10], and linear programming [32]. Its main advantage over the synchronous parallel optimization is avoiding the synchronization cost, so it minimizes the system overheads and maximizes the efficiency of all computation workers.

One of core theoretical questions is how much speedup (or benefit) the asynchronous parallelization can bring to us, that is, how much time can we save by employing more computation resources? More precisely, people are interested in the **r**unning **t**ime **s**peedup (RTS) with $T$ workers:

$$\text{RTS}(T) = \frac{\text{running time using a single worker}}{\text{running time using } T \text{ workers}}.$$

Since in the asynchronous parallelism all workers keep busy, RTS can be measured roughly by the **c**omputational **c**omplexity **s**peedup (CCS) with $T$ workers[1]

$$\text{CCS}(T) = \frac{\text{total computational complexity using a single worker}}{\text{total computational complexity using } T \text{ workers}} \times T.$$

In this paper, we are mainly interested in the conditions to ensure the linear speedup property. More specifically, what is the upper bound on $T$ to ensure $\text{CCS}(T) = \Theta(T)$?

Existing studies on special cases, such as asynchronous stochastic gradient descent (ASGD) and asynchronous stochastic coordinate descent (ASCD), have revealed some clues for what factors can

Table 1: Asynchronous parallel algorithms. "I" and "C" in "model" stand for inconsistent and consistent read model respectively, which will be explained later. "Base alg." is short for base algorithm.

| Asyn. alg. | base alg. | problem type | upper bound of $T$ | model |
|---|---|---|---|---|
| ASGD [25] | SGD | smooth, strongly convex | $O(N^{1/4})$ | C |
| ASGD [1] | SGD | smooth, convex | $O(K^{1/4}\min\{\sigma^{3/2},\sigma^{1/2}\})$ | C |
| ASGD[11] | SGD | composite, convex | $O(K^{1/4}\sigma^{1/2})$ | C |
| ASGD [18] | SGD | smooth, nonconvex | $O(N^{1/4}K^{1/2}\sigma)$ | I |
| ASGD [18] | SGD | smooth, nonconvex | $O(K^{1/2}\sigma)$ | C |
| ARK [21] | SGD | $Ax=b$ | $O(N)$ | C |
| ASCD [20] | SCD | smooth, convex, unconstrained | $O(N^{1/2})$ | C |
| ASCD [20] | SCD | smooth, convex, constrained | $O(N^{1/4})$ | C |
| ASCD [19] | SCD | composite, convex | $O(N^{1/4})$ | I |
| ASCD [3] | SCD | $\frac{1}{2}x^TAx - b^Tx$ | $O(N)$ | C |
| ASCD [3] | SCD | $\frac{1}{2}x^TAx - b^Tx$ | $O(N^{1/2})$ | I |
| ASCD [15] | SCD | $\frac{1}{2}x^TAx - b^Tx$ , constrained | $O(N^{1/2})$ | I |
| ASZD | zeroth order SGD & SCD | smooth, nonconvex | $O(\sqrt{N^{3/2}+KN^{1/2}\sigma^2})$ | I |
| ASGD | SGD | smooth, nonconvex | $O(\sqrt{N^{3/2}+KN^{1/2}\sigma^2})$ | I |
| ASGD | SGD | smooth, nonconvex | $O(\sqrt{K\sigma^2+1})$ | C |
| ASCD | SCD | smooth, nonconvex | $O(N^{3/4})$ | I |

affect the upper bound of $T$. For example, Agarwal and Duchi [1] showed the upper bound depends on the variance of the stochastic gradient in ASGD; Niu et al. [25] showed that the upper bound depends on the data sparsity and the dimension of the problem in ASGD; and Avron et al. [3], Liu and Wright [19] found that the upper bound depends on the problem dimension as well as the diagonal dominance of the Hessian matrix of the objective. However, it still lacks a comprehensive and generic analysis to comprehend all pieces and show how these factors jointly affect the speedup property.

This paper provides a comprehensive and generic analysis to study the speedup property for a broad range of asynchronous parallel stochastic algorithms from the zeroth order to the first order methods.

To avoid unnecessary complication and cover practical problems and algorithms, we consider the following nonconvex stochastic optimization problem:

$$\min_{x\in\mathbb{R}^N}\ f(x):=\mathbb{E}_\xi(F(x;\xi)), \tag{1}$$

where $\xi\in\Xi$ is a random variable, and both $F(\cdot;\xi):\mathbb{R}^N\to\mathbb{R}$ and $f(\cdot):\mathbb{R}^N\to\mathbb{R}$ are smooth but not necessarily convex functions. This objective function covers a large scope of machine learning problems including deep learning. $F(\cdot;\xi)$'s are called *component function*s in this paper. The most common specification is that $\Xi$ is an index set of all training samples $\Xi=\{1,2,\cdots,n\}$ and $F(x;\xi)$ is the loss function with respect to the training sample indexed by $\xi$.

We highlight the main contributions of this paper in the following:

- We provide a generic analysis for convergence and speedup, which covers many existing algorithms including ASCD, ASGD ( implementation on parameter server), ASGD (implementation on multicore systems), and others as its special cases.
- Our generic analysis can recover or improve the existing results on special cases.
- Our generic analysis suggests a novel asynchronous stochastic zeroth-order gradient descent (ASZD) algorithm and provides the analysis for its convergence rate and speedup property. To the best of our knowledge, this is the first asynchronous parallel *zeroth* order algorithm.
- The experiment includes a novel application of the proposed ASZD method on model blending and hyper parameter tuning for big data optimization.

## 1.1 Related Works

We first review *first-order asynchronous parallel stochastic algorithms*. Table 1 summarizes existing linear speedup results for asynchronous parallel optimization algorithms mostly related to this paper. The last block of Table 1 shows the results in this paper. Reddi et al. [29] proved the convergence of asynchronous variance reduced stochastic gradient (SVRG) method and its speedup in sparse setting. Mania et al. [22] provides a general perspective (or starting point) to analyze for asynchronous stochastic algorithms, including HOGWILD!, asynchronous SCD and asynchronous sparse SVRG. The fundamental difference in our work lies on that we apply different analysis and our result can be

directly applied to various special cases, while theirs cannot. In addition, there is a line of research studying the asynchronous ADMM type methods, which is not in the scope of this paper. We encourage readers to refer to recent literatures, for example, Hong [14], Zhang and Kwok [35].

We end this section by reviewing the zeroth-order stochastic methods. We use $N$ to denote the dimension of the problem, $K$ to denote the iteration number, and $\sigma$ to the variance of stochastic gradient. Nesterov and Spokoiny [24] proved a convergence rate of $O(N/\sqrt{K})$ for zeroth-order SGD applied to convex optimization. Based on [24], Ghadimi and Lan [12] proved a convergence rate of $O(\sqrt{N/K})$ rate for zeroth-order SGD on nonconvex smooth problems. Jamieson et al. [16] shows a lower bound $O(1/\sqrt{K})$ for any zeroth-order method with inaccurate evaluation. Duchi et al. [9] proved a $O(N^{1/4}/K + 1/\sqrt{K})$ rate for zeroth order SGD on convex objectives but with some very different assumptions compared to our paper. Agarwal et al. [2] proved a regret of $O(\text{poly}(N)\sqrt{K})$ for zeroth-order bandit algorithm on convex objectives.

For more comprehensive review of asynchronous algorithms, please refer to the long version of this paper on arXiv:1606.00498.

## 1.2 Notation

- $e_i \in \mathbb{R}^N$ denotes the $i^{\text{th}}$ natural unit basis vector.
- $\mathbb{E}(\cdot)$ means taking the expectation with respect to all random variables, while $\mathbb{E}_a(\cdot)$ denotes the expectation with respect to a random variable $a$.
- $\nabla f(x) \in \mathbb{R}^N$ is the gradient of $f(x)$ with respect to $x$. Let $S$ be a subset of $\{1, \cdots, N\}$. $\nabla_S f(x) \in \mathbb{R}^N$ is the projection of $\nabla f(x)$ onto the index set $S$, that is, setting components of $\nabla f(x)$ outside of $S$ to be zero. We use $\nabla_i f(x) \in \mathbb{R}^N$ to denote $\nabla_{\{i\}} f(x)$ for short.
- $f^*$ denotes the optimal objective value in (1).

## 2 Algorithm

We illustrate the asynchronous parallelism by assuming a centralized network: a central node and multiple child nodes (workers). The central node maintains the optimization variable $x$. It could be a parameter server if implemented on a computer cluster [17]; it could be a shared memory if implemented on a multicore machine. Given a base algorithm $\mathcal{A}$, all child nodes run algorithm $\mathcal{A}$ independently and con-

---

**Algorithm 1 G**eneric **A**synchronous **S**tochastic **A**lgorithm (GASA)

---

**Require:** $x_0, K, Y, (\mu_1, \mu_2, \ldots, \mu_N), \{\gamma_k\}_{k=0,\ldots,K-1}$ ▷ $\gamma_k$ is the step length for $k^{\text{th}}$ iteration
**Ensure:** $\{x_k\}_{k=0}^{K}$
 1: **for** $k = 0, \ldots, K - 1$ **do**
 2:     Randomly select a component function index $\xi_k$ and a set of coordinate indices $S_k$, where $|S_k| = Y$;
 3:     $x_{k+1} = x_k - \gamma_k G_{S_k}(\hat{x}_k; \xi_k)$;
 4: **end for**

---

currently: read $x$ from the central node (we call the result of this read $\hat{x}$, and it is mathematically defined later in (4)), calculate locally using the $\hat{x}$, and modify $x$ on the central node. There is no need to synchronize child nodes. Therefore, all child nodes stay busy and consequently their efficiency gets maximized. In other words, we have $\text{CCS}(T) \approx \text{RTS}(T)$. Note that due to the asynchronous parallel mechanism the variable $x$ in the central node is not updated exactly following the protocol of Algorithm $\mathcal{A}$, since when a child node returns its computation result, the $x$ in the central node might have been changed by other child nodes. Thus a new analysis is required. A fundamental question would be under what conditions a linear speedup can be guaranteed. In other words, under what conditions $\text{CCS}(T) = \Theta(T)$ or equivalently $\text{RTS}(T) = \Theta(T)$?

To provide a comprehensive analysis, we consider a generic algorithm $\mathcal{A}$ – the zeroth order hybrid of SCD and SGD: iteratively sample a component function[2] indexed by $\xi$ and a coordinate block $S \subseteq \{1, 2, \cdots, N\}$, where $|S| = Y$ for some constant $Y$ and update $x$ with

$$x \leftarrow x - \gamma G_S(x; \xi) \qquad (2)$$

where $G_S(x; \xi)$ is an approximation to the block coordinate stochastic gradient $NY^{-1}\nabla_S F(x; \xi)$:

$$G_S(x; \xi) := \sum_{i \in S} \frac{N}{2Y\mu_i}(F(x + \mu_i e_i; \xi) - F(x - \mu_i e_i; \xi))e_i, \quad S \subseteq \{1, 2, \ldots, N\}. \qquad (3)$$

In the definition of $G_S(x; \xi)$, $\mu_i$ is the approximation parameter for the $i^{\text{th}}$ coordinate. $(\mu_1, \mu_2, \ldots, \mu_N)$ is predefined in practice. We only use the function value (the zeroth order information) to estimate $G_S(x; \xi)$. It is easy to see that the closer to 0 the $\mu_i$'s are, the closer $G_S(x; \xi)$ and $NY^{-1}\nabla_S f(x; \xi)$ will be. In particular, $\lim_{\mu_i \to 0, \forall i} G_S(x; \xi) = NY^{-1}\nabla_S f(x; \xi)$.

Applying the asynchronous parallelism, we propose a generic asynchronous stochastic algorithm in Algorithm 1. This algorithm essentially characterizes how the value of $x$ is updated in the central node. $\gamma_k$ is the predefined steplength (or learning rate). $K$ is the total number of iterations (note that this iteration number is counted by the the central node, that is, any update on $x$ no matter from which child node will increase this counter.)

As we mentioned, the key difference of the asynchronous algorithm from the protocol of Algorithm $\mathcal{A}$ in Eq. (2) is that $\hat{x}_k$ may be not equal to $x_k$. In asynchronous parallelism, there are two different ways to model the value of $\hat{x}_k$:

• **Consistent read:** $\hat{x}_k$ is some early *existed* state of $x$ in the central node, that is, $\hat{x}_k = x_{k-\tau_k}$ for some $\tau_k \geq 0$. This happens if reading $x$ and writing $x$ on the central node by any child node are atomic operations, for instance, the implementation on a parameter server [17].

• **Inconsistent read:** $\hat{x}_k$ could be more complicated when the atomic read on $x$ cannot be guaranteed, which could happen, for example, in the implementation on the multi-core system. It means that while one child is reading $x$ in the central node, other child nodes may be performing modifications on $x$ at the same time. Therefore, different coordinates of $x$ read by any child node may have different ages. In other words, $\hat{x}_k$ may not be any *existed* state of $x$ in the central node.

Readers who want to learn more details about consistent read and inconsistent read can refer to [3, 18, 19]. To cover both cases, we note that $\hat{x}_k$ can be represented in the following generic form:

$$\hat{x}_k = x_k - \sum_{j \in J(k)} (x_{j+1} - x_j), \qquad (4)$$

where $J(k) \subset \{k-1, k-2, \ldots, k-T\}$ is a subset of the indices of early iterations, and $T$ is the upper bound for staleness. This expression is also considered in [3, 18, 19, 27]. *Note that the practical value of $T$ is usually proportional to the number of involved nodes (or workers).* Therefore, the total number of workers and the upper bound of the staleness are treated as the same in the following discussion and this notation $T$ is abused for simplicity.

## 3   Theoretical Analysis

Before we show the main results of this paper, let us first make some global assumptions commonly used for the analysis of stochastic algorithms.[3]

**Bounded Variance of Stochastic Gradient** $\mathbb{E}_\xi(\|\nabla F(x;\xi) - \nabla f(x)\|^2) \leq \sigma^2, \forall x.$

**Lipschitzian Gradient** The gradient of both the objective and its component functions are Lipschitzian:[4]

$$\max\{\|\nabla f(x) - \nabla f(y)\|, \|\nabla F(x;\xi) - \nabla F(y;\xi)\|\} \leq L\|x - y\| \quad \forall x, \forall y, \forall \xi. \qquad (5)$$

Under the Lipschitzian gradient assumption, define two more constants $L_s$ and $L_{\max}$. Let $s$ be any positive integer bounded by $N$. Define $L_s$ to be the minimal constant satisfying the following inequality: $\forall \xi, \forall x, \alpha_i e_i \forall S \subset \{1, 2, ..., N\}$ with $|S| \leq s$ for any $z = \sum_{i \in S}$ we have:

$$\max\{\|\nabla f(x) - \nabla f(x + z)\|, \|\nabla F(x;\xi) - \nabla F(x + z;\xi)\|\} \leq L_s\|z\|$$

Define $L_{(i)}$ for $i \in \{1, 2, \ldots, N\}$ as the minimum constant that satisfies:

$$\max\{\|\nabla_i f(x) - \nabla_i f(x + \alpha e_i)\|, \|\nabla_i F(x;\xi) - \nabla_i F(x + \alpha e_i;\xi)\|\} \leq L_{(i)}|\alpha|. \quad \forall \xi, \forall x. \qquad (6)$$

Define $L_{\max} := \max_{i \in \{1, \ldots, N\}} L_{(i)}$. It can be seen that $L_{\max} \leq L_s \leq L$.

**Independence** All random variables $\xi_k, S_k$ for $k = 0, 1, \cdots, K$ are independent to each other.

**Bounded Age** Let $T$ be the global bound for delay: $J(k) \subseteq \{k - 1, \ldots, k - T\}, \forall k$, so $|J(k)| \leq T$.

We define the following global quantities for short notations:

$$\omega := \left(\sum_{i=1}^N L_{(i)}^2 \mu_i^2\right)/N, \quad \alpha_1 := 4 + 4\left(TY + Y^{3/2}T^2/\sqrt{N}\right)L_T^2/(L_Y^2 N),$$

$$\alpha_2 := Y/((f(x_0) - f^*)L_Y N), \quad \alpha_3 := (K(N\omega + \sigma^2)\alpha_2 + 4)L_Y^2/L_T^2. \qquad (7)$$

Next we show our main result in the following theorem:

**Theorem 1** (Generic Convergence Rate for GASA). *Choose the steplength $\gamma_k$ to be a constant $\gamma$ in Algorithm 1*

$$\gamma_k^{-1} = \gamma^{-1} = 2L_Y NY^{-1}\left(\sqrt{\alpha_1^2/(K(N\omega + \sigma^2)\alpha_2 + \alpha_1)} + \sqrt{K(N\omega + \sigma^2)\alpha_2}\right), \forall k$$

*and suppose the age $T$ is bounded by $T \le \frac{\sqrt{N}}{2Y^{1/2}}\left(\sqrt{1 + 4Y^{-1/2}N^{1/2}\alpha_3} - 1\right)$. We have the following convergence rate:*

$$\frac{\sum_{k=0}^{K}\mathbb{E}\|\nabla f(x_k)\|^2}{K} \le \frac{20}{K\alpha_2} + \frac{1}{K\alpha_2}\left(\frac{L_T^2}{L_Y^2}\frac{\sqrt{1 + 4Y^{-1/2}N^{1/2}\alpha_3} - 1}{\sqrt{NY^{-1}}} + 11\sqrt{N\omega + \sigma^2}\sqrt{K\alpha_2}\right) + N\omega. \tag{8}$$

Roughly speaking, the first term on the RHS of (8) is related to SCD; the second term is related to "stochastic" gradient descent; and the last term is due to the zeroth-order approximation.

Although this result looks complicated (or may be less elegant), it is capable to capture many important subtle structures, which can be seen by the subsequent discussion. We will show how to recover and improve existing results as well as prove the convergence for new algorithms using Theorem 1. To make the results more interpretable, we use the big-O notation to avoid explicitly writing down all the constant factors, including all $L$'s, $f(x_0)$, and $f^*$ in the following corollaries.

### 3.1 **A**synchronous **S**tochastic **C**oordinate **D**escent (ASCD)

We apply Theorem 1 to study the asynchronous SCD algorithm by taking $Y = 1$ and $\sigma = 0$. $S_k = \{i_k\}$ only contains a single randomly sampled coordinate, and $\omega = 0$ (or equivalently $\mu_i = 0, \forall i$). The essential updating rule on $x$ is $x_{k+1} = x_k - \gamma_k \nabla_{i_k} f(\hat{x}_k)$.

**Corollary 2** (ASCD). *Let $\omega = 0$, $\sigma = 0$, and $Y = 1$ in Algorithm 1 and Theorem 1. If*

$$T \le O(N^{3/4}), \tag{9}$$

*the following convergence rate holds:*

$$\left(\sum_{k=0}^{K}\mathbb{E}\|\nabla f(x_k)\|^2\right)/K \le O(N/K). \tag{10}$$

The proved convergence rate $O(N/K)$ is consistent with the existing analysis of SCD [30] or ASCD for smooth optimization [20]. However, our requirement in (9) to ensure the linear speedup property is better than the one in [20], by improving it from $T \le O(N^{1/2})$ to $T \le O(N^{3/4})$. Mania et al. [22] analyzed ASCD for strongly convex objectives and proved a linear speedup smaller than $O(N^{1/6})$, which is also more restrictive than ours.

### 3.2 **A**synchronous **S**tochastic **G**radient **D**escent (ASGD)

ASGD has been widely used to solve deep learning [7, 26, 36], NLP [4, 13], and many other important machine learning problems [25]. There are two typical implementations of ASGD. The first type is to implement on the computer cluster with a parameter sever [1, 17]. The parameter server serves as the central node. It can ensure the atomic read or write of the whole vector $x$ and leads to the following updating rule for $x$ (setting $Y = N$ and $\mu_i = 0, \forall i$ in Algorithm 1):

$$x_{k+1} = x_k - \gamma_k \nabla F(\hat{x}_k; \xi_k). \tag{11}$$

Note that a single iteration is defined as modifying the whole vector. The other type is to implement on a single computer with multiple cores. In this case, the central node corresponds to the shared memory. Multiple cores (or threads) can access it simultaneously. However, in this model atomic read and write of $x$ cannot be guaranteed. Therefore, for the purpose of analysis, each update on a single coordinate accounts for an iteration. It turns out to be the following updating rule (setting $S_k = \{i_k\}$, that is, $Y = 1$, and $\mu_i = 0, \forall i$ in Algorithm 1):

$$x_{k+1} = x_k - \gamma_k \nabla_{i_k} F(\hat{x}_k; \xi_k). \tag{12}$$

Readers can refer to [3, 18, 25] for more details and illustrations for these two implementations.

**Corollary 3** (ASGD in (11)). *Let $\omega = 0$ (or $\mu_i = 0, \forall i$ equivalently) and $Y = N$ in Algorithm 1 and Theorem 1. If*

$$T \le O\left(\sqrt{K\sigma^2 + 1}\right), \tag{13}$$

*then the following convergence rate holds:*

$$\left(\sum_{k=0}^{K}\mathbb{E}\|\nabla f(x_k)\|^2\right)/K \le O\left(\sigma/\sqrt{K} + 1/K\right). \tag{14}$$

First note that the convergence rate in (14) is tight since it is consistent with the serial (nonparallel) version of SGD [23]. We compare this linear speedup property indicated by (13) with results in [1], [11], and [18]. To ensure such rate, Agarwal and Duchi [1] need $T$ to be bounded by $T \leq O(K^{1/4} \min\{\sigma^{3/2}, \sqrt{\sigma}\})$, which is inferior to our result in (13). Feyzmahdavian et al. [11] need $T$ to be bounded by $\sigma^{1/2} K^{1/4}$ to achieve the same rate, which is also inferior to our result. Our requirement is consistent with the one in [18]. To the best of our knowledge, it is the best result so far.

**Corollary 4** (ASGD in (12)). *Let $\omega = 0$ (or equivalently, $\mu_i = 0, \forall i$) and $Y = 1$ in Algorithm 1 and Theorem 1. If*

$$T \leqslant O\left(\sqrt{N^{3/2} + KN^{1/2}\sigma^2}\right), \tag{15}$$

*then the following convergence rate holds*

$$\left(\textstyle\sum_{k=0}^{K} \mathbb{E}\|\nabla f(x_k)\|^2\right)/K \leqslant O\left(\sqrt{N/K}\sigma + N/K\right). \tag{16}$$

The additional factor $N$ in (16) (comparing to (14)) arises from the different way of counting the iteration. This additional factor also appears in [25] and [18]. We first compare our result with [18], which requires $T$ to be bounded by $O(\sqrt{KN^{1/2}\sigma^2})$. We can see that our requirement in (16) allows a larger value for $T$, especially when $\sigma$ is small such that $N^{3/2}$ dominates $KN^{1/2}\sigma^2$. Next we compare with [25], which assumes that the objective function is strongly convex. Although this is sort of comparing "apple" with "orange", it is still meaningful if one believes that the strong convexity would not affect the linear speedup property, which is implied by [22]. In [25], the linear speedup is guaranteed if $T \leq O(N^{1/4})$ under the assumption that the sparsity of the stochastic gradient is bounded by $O(1)$. In comparison, we do not require the assumption of sparsity for stochastic gradient and have a better dependence on $N$. Moreover, beyond the improvement over existing analysis in [22] and [18], our analysis provides some interesting insights for asynchronous parallelism. Niu et al. [25] essentially suggests a large problem dimension $N$ is beneficial to the linear speedup, while Lian et al. [18] and many others (for example, Agarwal and Duchi [1], Feyzmahdavian et al. [11]) suggest that a large stochastic variance $\sigma$ (this often implies the number of samples is large) is beneficial to the linear speedup. Our analysis shows the combo effect of $N$ and $\sigma$ and shows how they improve the linear speedup jointly.

### 3.3 **A**synchronous **S**tochastic **Z**eroth-order **D**escent (ASZD)

We end this section by applying Theorem 1 to generate a novel asynchronous zeroth-order stochastic descent algorithm, by setting the block size $Y = 1$ (or equivalently $S_k = \{i_k\}$) in $G_{S_k}(\hat{x}_k; \xi_k)$

$$G_{S_k}(\hat{x}_k; \xi_k) = G_{\{i_k\}}(\hat{x}_k; \xi_k) = (F(\hat{x}_k + \mu_{i_k} e_{i_k}; \xi_k) - F(\hat{x}_k - \mu_{i_k} e_{i_k}; \xi_k))/(2\mu_{i_k})e_{i_k}. \tag{17}$$

To the best of our knowledge, this is the first asynchronous algorithm for zeroth-order optimization.

**Corollary 5** (ASZD). *Set $Y = 1$ and all $\mu_i$'s to be a constant $\mu$ in Algorithm 1. Suppose that $\mu$ satisfies*

$$\mu \leqslant O\left(1/\sqrt{K} + \min\left\{\sqrt{\sigma}(NK)^{-1/4}, \sigma/\sqrt{N}\right\}\right), \tag{18}$$

*and $T$ satisfies*

$$T \leqslant O\left(\sqrt{N^{3/2} + KN^{1/2}\sigma^2}\right). \tag{19}$$

*We have the following convergence rate*

$$\left(\textstyle\sum_{k=0}^{K} \mathbb{E}\|\nabla f(x_k)\|^2\right)/K \leqslant O\left(N/K + \sqrt{N/K}\sigma\right). \tag{20}$$

We firstly note that the convergence rate in (20) is consistent with the rate for the serial (nonparallel) zeroth-order stochastic gradient method in [12]. Then we evaluate this result from two perspectives.

First, we consider $T = 1$, which leads to the serial (non-parallel) zeroth-order stochastic descent. Our result implies a better dependence on $\mu$, comparing with [12].[5] To obtain such convergence rate

in (20), Ghadimi and Lan [12] require $\mu \leqslant O\left(1/(N\sqrt{K})\right)$, while our requirement in (18) is much less restrictive. An important insight in our requirement is to suggest the dependence on the variance $\sigma$: if the variance $\sigma$ is large, $\mu$ is allowed to be a much larger value. This insight meets the common sense: a large variance means that the stochastic gradient may largely deviate from the true gradient, so we are allowed to choose a large $\mu$ to obtain a less exact estimation for the stochastic gradient without affecting the convergence rate. From the practical view of point, it always tends to choose a large value for $\mu$. Recall the zeroth-order method uses the function difference at two different points (e.g., $x + \mu e_i$ and $x - \mu e_i$) to estimate the differential. In a practical system (e.g., a concrete control system), there usually exists some system noise while querying the function values. If two points are too close (in other words $\mu$ is too small), the obtained function difference is dominated by noise and does not really reflect the function differential.

Second, we consider the case $T \geq 1$, which leads to the asynchronous zeroth-order stochastic descent. To the best of our knowledge, this is the first such algorithm. The upper bound for $T$ in (19) essentially indicates the requirement for the linear speedup property. The linear speedup property here also shows that even if $K\sigma^2$ is much smaller than 1, we still have $O(N^{3/4})$ linear speedup, which shows a fundamental understanding of asynchronous stochastic algorithms that $N$ and $\sigma$ can improve the linear speedup jointly.

## 4 Experiment

Since the ASCD and various ASGDs have been extensively validated in recent papers. We conduct two experiments to validate the proposed ASZD on in this section. The first part applies ASZD to estimate the parameters for a synthetic black box system. The second part applies ASZD to the model combination for Yahoo Music Recommendation Competition.

### 4.1 Parameter Optimization for A Black Box

We use a deep neural network to simulate a black box system. The optimization variables are the weights associated with a neural network. We choose 5 layers ($400/100/50/20/10$ nodes) for the neural network with $46380$ weights (or parameters) totally. The weights are randomly generated from i.i.d. Gaussian distribution. The output vector is constructed by applying the network to the input vector plus some Gaussian random noise. We use this network to generate $463800$ samples. These synthetic samples are used to optimize the weights for the black box. (We pretend not to know the structure and weights of this neural network because it is a black box.) To optimize (estimate) the parameters for this black box, we apply the proposed ASZD method.

The experiment is conducted on the machine (Intel Xeon architecture), which has 4 sockets and 10 cores for each socket. We run Algorithm 1 on various numbers of cores from 1 to 32 and the steplength is chosen as $\gamma = 0.1$, which is based on the best performance of Algorithm 1 running on 1 core to achieve the precision $10^{-1}$ for the objective value.

Table 2: CCR and RTS of ASZD for different # of threads (synthetic data).

| thr-# | 1 | 4 | 8 | 12 | 16 | 20 | 24 | 28 | 32 |
|-------|---|-----|------|------|-------|-------|-------|-------|-------|
| CCS | 1 | 3.87 | 7.91 | 9.97 | 14.74 | 17.86 | 21.76 | 26.44 | 30.86 |
| RTS | 1 | 3.32 | 6.74 | 8.48 | 12.49 | 15.08 | 18.52 | 22.49 | 26.12 |

The speedup is reported in Table 2. We observe that the iteration speedup is almost linear while the running time speedup is slightly worse than the iteration speedup. We also draw Figure 1 (see the supplement) to show the curve of the objective value against the number of iterations and running time respectively.

### 4.2 Asynchronous Parallel Model Combination for Yahoo Music Recommendation Competition

In KDD-Cup 2011, teams were challenged to predict user ratings in music given the Yahoo! Music data set [8]. The evaluation criterion is the Root Mean Squared Error (RMSE) of the test data set:

$$\text{RMSE} = \sqrt{\textstyle\sum_{(u,i)\in\mathcal{T}_1}(r_{ui} - \hat{r}_{ui})^2/|\mathcal{T}_1|}, \tag{21}$$

where $(u,i) \in \mathcal{T}_1$ are all user ratings in Track 1 test data set (6,005,940 ratings), $r_{ui}$ is the true rating for user $u$ and item $i$, and $\hat{r}_{ui}$ is the predicted rating. The winning team from NTU created more than 200 models using different machine learning algorithms [6], including Matrix Factorization, k-NN, Restricted Boltzmann Machines, etc. They blend these models using Neural Network and Binned Linear Regression on the validation data set (4,003,960 ratings) to create a model ensemble to achieve better RMSE.

We were able to obtain the predicted ratings of $N = 237$ individual models on the KDD-Cup test data set from the NTU KDD-Cup team, which is a matrix $X$ with 6,005,940 rows (corresponding to the 6,005,940 test data set samples) and 237 columns. Each element $X_{ij}$ indicates the $j$-th model's predicted rating on the $i$-th Yahoo! Music test data sample. In our experiments, we try to linearly blend the 237 models using information from the test data set. Thus, our variable to optimize is a vector $x \in \mathbb{R}^N$ as coefficients of the predicted ratings for each model. To ensure that our linear blending does not over-fit, we further split $X$ randomly into two equal parts, calling them the "validation" set (denoted as $A \in \mathbb{R}^{n \times N}$) for model blending and the true test set.

We define our objective function as RMSE$^2$ of the blended output on the validation set: $f(x) = \|Ax - r\|^2/n$ where $r$ is the corresponding true ratings in the validation set and $Ax$ is the predicted ratings after blending.

We assume that we cannot see the entries of $r$ directly, and thus cannot compute the gradient of $f(x)$. In our experiment, we treat $f(x)$ as a blackbox, and the only information we can get from it is its value given a model blending coefficients $x$. This is similar to submitting a model for KDD-Cup and obtain a leader-board RMSE of the test set; we do not know the actual values of the test set. Then, we apply our ASZD algorithm to minimize $f(x)$ with zero-order information only.

Table 3: Comparing RMSEs on test data set with KDD-Cup winner teams

|  | NTU (1st) | Commendo (2nd) | InnerPeace (3rd) | Our result |
|---|---|---|---|---|
| RMSE | 21.0004 | 21.0545 | 21.2335 | 21.1241 |

We implement our algorithm using Julia on a 10-core Xeon E7-4680 machine an run our algorithm for the same number of iterations, with different number of threads, and measured the running time speedup (RTS) in Figure 4 (see supplement). Similar to our experiment on neural network blackbox, our algorithm has a almost linear speedup. For completeness, Figure 2 in supplement shows the square root of objective function value (RMSE) against the number of iterations and running time. After about 150 seconds, our algorithm running with 10 threads achieves a RMSE of 21.1241 on our test set. Our results are comparable to KDD-Cup winners, as shown in Table 3. Since our goal is to show the performance of our algorithm, we assume we can "submit" our solution $x$ for unlimited times, which is unreal in a real contest like KDD-Cup. However, even with very few iterations, our algorithm does converge fast to a reasonable small RMSE, as shown in Figure 3.

## 5 Conclusion

In this paper, we provide a generic linear speedup analysis for the zeroth-order and first-order asynchronous parallel algorithms. Our generic analysis can recover or improve the existing results on special cases, such as ASCD, ASGD (parameter implementation), ASGD (multicore implementation). Our generic analysis also suggests a novel ASZD algorithm with guaranteed convergence rate and speedup property. To the best of our knowledge, this is the first asynchronous parallel *zeroth* order algorithm. The experiment includes a novel application of the proposed ASZD method on model blending and hyper parameter tuning for big data optimization.

## Acknowledgements

This project is in part supported by the NSF grant CNS-1548078. We especially thank Chen-Tse Tsai for providing the code and data for the Yahoo Music Competition.

## Footnotes

[1]For simplicity, we assume that the communication cost is not dominant throughout this paper.

[2]The algorithm and theoretical analysis followed can be easily extended to the minibatch version.

[3]Some underlying assumptions such as reading and writing a float number are omitted here. As pointed in [25], these behaviors are guaranteed by most modern architectures.

[4]Note that the Lipschitz assumption on the component function $F(x;\xi)$'s can be eliminated when it comes to first order methods (i.e., $\omega \to 0$) in our following theorems.

[5] Acute readers may notice that our way in (17) to estimate the stochastic gradient is different from the one used in [12]. Our method only estimates a single coordinate gradient of a sampled component function, while Ghadimi and Lan [12] estimate the whole gradient of the sampled component function. Our estimation is more accurate but less aggressive. The proved convergence rate actually improves a small constant in [12].

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
