[Supplementary Material · nips_supp.pdf]

# Appendix: Supplemental Figures for Experiment

Figure 1: Deep neural network for synthetic data using Algorithm 1. The Algorithm 1 algorithm is run on various numbers of machines from 1 to 32. The curves of the objective loss against the number of iteration and the running time are drawn in the left and the right graphs respectively.

(a) RMSE on validation set w.r.t iteration

(b) RMSE on validation set w.r.t time

(c) RMSE on test set w.r.t iteration

(d) RMSE on test set w.r.t time

Figure 2: Model blending with Yahoo! Music test data set using ASZD.

Figure 3: We zoom in for the first a few thousands iterations in Figure 2. Our algorithm quickly approaches to a reasonable RMSE.

Figure 4: Running time speedup of Algorithm 1 is almost linear. On a 10-core machine we can achieve a 8x speedup.

# Supplemental Materials for Proofs

We first show some preliminary results about the dependence between the zeroth-order gradient and the true gradient, and then prove the convergence property for GASA (Algorithm 1).

## .1 Preliminary results

Given a function $p(x) : \mathbb{R}^N \to \mathbb{R}$ and a predefined approximation parameter vector $(\mu_1, \mu_2, \cdots, \mu_N)^\top \in \mathbb{R}_{++}^N$, we define the smoothing function of $p(x)$ with respect to the $i$th dimension:

$$p_i(x) := \mathbb{E}_{\{v \sim U_{[-\mu_i, \mu_i]}\}}(p(x + v e_i)) = \frac{1}{2\mu_i} \int_{-\mu_i}^{\mu_i} p(x + v e_i) dv = \frac{1}{2} \int_{-1}^{1} p(x + v \mu_i e_i) dv. \quad (22)$$

where $v \sim U_{[-\mu_i, \mu_i]}$ means that $v$ follows the uniform distribution over the range $[-\mu_i, \mu_i]$. $\nabla_x$ and $\nabla_v$ denote taking gradient with respect to variable $x$ and $v$ respectively. $\nabla_i$ is short for $e_i e_i^\top \nabla_x$.

**Lemma 6** (Zeroth Order Approximation). *Suppose that function $p(x) : x \in \mathbb{R}^N \to \mathbb{R}$ has Lipschitzian gradient with parameter $L$ and $p(x + v e_i) : v \in \mathbb{R} \to \mathbb{R}$ has Lipschitzian gradient with parameter $L_{(i)}$ for all $x$. Given an approximation parameter vector $(\mu_1, \mu_2, \cdots, \mu_N)^\top \in \mathbb{R}_{++}^N$, we have:*

*1. The smoothing function $p_i$ is differentiable, and also has the Lipschitzian gradient with Lipschitz constant $L$.*
*2. For the $i^{th}$ coordinate,*

$$\nabla_i p_i(x) = \frac{1}{2\mu_i}(p(x + \mu_i e_i) - p(x - \mu_i e_i)) e_i.$$

*3. For any $x \in \mathbb{R}^N, i \in \{1, \dots, N\}$, we have*

$$|p_i(x) - p(x)| \quad \leqslant \quad \frac{L_{(i)} \mu_i^2}{2}, \quad (23)$$

$$\|\nabla_i p_i(x) - \nabla_i p(x)\| \quad \leqslant \quad \frac{L_{(i)} \mu_i}{2}, \quad (24)$$

$$\mathbb{E}_i \|\nabla_i p_i(x) - \nabla_i p(x)\|^2 \quad \leqslant \quad \frac{\omega}{4}, \quad (25)$$

*where $\omega$ is defined in (7).*
*4. If $p$ is convex, then $p_i$ is also convex.*

*Proof.* To prove the first statement, we note that

$$\nabla p_i(x) \quad = \quad \frac{1}{2} \int_{-1}^{1} \nabla_x p(x + \mu_i v e_i) dv.$$

Since function $p(x)$ has Lipschitzian gradient, we have

$$\begin{aligned}
\|\nabla p_i(x) - \nabla p_i(y)\| &= \|\mathbb{E}_{\{v \sim U_{[-1,1]}\}}(\nabla p(x + \mu_i v e_i) - \nabla p(y + \mu_i v e_i))\| \\
&\leqslant \mathbb{E}_{\{v \sim U_{[-1,1]}\}}(L\|x - y\|) \\
&= L\|x - y\|,
\end{aligned}$$

which implies the first statement.

To prove the second statement, we have

$$\begin{aligned}
\nabla_i p_i(x) &= \frac{1}{2} \left( \int_{-1}^{1} \nabla_i p(x + \mu_i v e_i) dv \right) \\
&= \frac{1}{2\mu_i} \left( \int_{-1}^{1} \nabla_v p(x + \mu_i v e_i) dv \right) e_i \\
&= \frac{1}{2\mu_i} (p(x + \mu_i e_i) - p(x - \mu_i e_i)) e_i,
\end{aligned}$$

where the last step comes from the Stokes' theorem.

Next we prove the third statement. First (23) can be proved by

$$
\begin{aligned}
|p_i(x) - p(x)| \;&\leqslant\; \frac{1}{2}\left|\int_{-1}^{1} p(x+\mu_i v e_i) - p(x+0\mu_i e_i)dv\right| \\
&=\; \frac{1}{2}\left|\int_{-1}^{1} p(x+\mu_i v e_i) - p(x+0\mu_i e_i) - \langle \nabla_v p(x), \mu_i v\rangle dv\right| \\
&\leqslant\; \frac{1}{2}\int_{-1}^{1} |p(x+\mu_i v e_i) - p(x) - \langle \nabla_v p(x), \mu_i v\rangle| dv \\
&\leqslant\; \frac{1}{2}\int_{-1}^{1} \frac{L_{(i)}\mu_i^2 v^2}{2} dv \\
&\leqslant\; \frac{L_{(i)}\mu_i^2}{2},
\end{aligned}
$$

where the second step is based on the observation

$$
\int_{-1}^{1} \langle \nabla_v p(x),\ \mu_i v\rangle dv = 0,
$$

and the second last step uses the assumption that $p(x+ve_i)$ has Lipschitzian gradient (with constant $L_{(i)}$) with respect to $v$. (24) can be proved from

$$
\begin{aligned}
\|\nabla_i p_i(x) - \nabla_i p(x)\| \;&=\; \left\|\frac{1}{2\mu_i}(p(x+\mu_i e_i) - p(x-\mu_i e_i))e_i - \nabla_i p(x)\right\| \\
&=\; \left\|\frac{1}{2\mu_i}((p(x+\mu_i e_i) - p(x-\mu_i e_i))e_i - 2\mu_i \nabla_i p(x))\right\| \\
&\leqslant\; \left\|\frac{1}{2\mu_i}((p(x+\mu_i e_i) - p(x))e_i - \mu_i \nabla_i p(x))\right\| \\
&\quad +\; \left\|\frac{1}{2\mu_i}((p(x) - p(x-\mu_i e_i))e_i - \mu_i \nabla_i p(x))\right\| \\
&\leqslant\; \frac{1}{2\mu_i}L_{(i)}\mu_i^2 = \frac{L_{(i)}\mu_i}{2}.
\end{aligned}
$$

(25) can be proved by

$$
\mathbb{E}_i\|\nabla_i p_i(x) - \nabla_i p(x)\|^2 \;\overset{(24)}{\leqslant}\; \sum_i \frac{L_{(i)}^2\mu_i^2}{4N} = \frac{\omega}{4}.
$$

For the last statement, given any $\theta \in [0,1]$, we have

$$
\begin{aligned}
\theta p_i(x) + (1-\theta)p_i(y) \;&=\; \frac{1}{2}\int_{-1}^{1} \theta p(x+\mu_i v e_i) + (1-\theta)p(y+\mu_i v e_i)dv \\
&\geqslant\; \frac{1}{2}\int_{-1}^{1} p(\theta x + (1-\theta)y + \mu_i v e_i)dv \\
&=\; p_i(\theta x + (1-\theta)y).
\end{aligned}
$$

It proves the convexity of $p_i(x)$. □

### .2 Proofs to Theorem 1
We first prove the general convergence property for Algorithm 1 and then apply it to prove Theorem 1.

**Theorem 7** (Convergence). *If the stepsize $\gamma_k$ in Algorithm 1 is appropriately chosen to satisfy*

$$
\Theta_k := \frac{N\gamma_k}{2} - 2\gamma_k^2\frac{L_Y}{Y}N^2 - 2L_T^2\frac{N^2}{Y^2}\gamma_k \sum_{\nu=1}^{T}\gamma_{k+\nu}\left(\gamma_k Y + \frac{Y^{3/2}\sum_{j'\in J(k+\nu)\setminus\{k\}}\gamma_{j'}}{\sqrt{N}}\right) \geqslant 0, \forall k,
$$

(26)

*we have*

$$\frac{1}{2}\sum_{k=0}^{K}\gamma_k\mathbb{E}\|\nabla f(x_k)\|^2 \quad\leqslant\quad f(x_0) - f^* + 2N\frac{L_T^2}{Y}\left(\frac{3N\omega}{2} + 3\sigma^2\right)\sum_{k=0}^{K}\left(\gamma_k\sum_{j\in J(k)}\gamma_j^2\right)$$

$$+ \left(\frac{L_Y}{Y}N\sigma^2 + \frac{L_Y}{Y}N^2\omega\right)\sum_{k=0}^{K}\gamma_k^2 + \frac{1}{4}N\omega\sum_{k=0}^{K}\gamma_k.$$

*Proof.* Besides of (3), we introduce one more notation here to abbreviate the notation:

$$G_i(x;\xi) = \frac{N}{2\mu_i}(F(x+\mu_i e_i;\xi) - F(x-\mu_i e_i;\xi))e_i.$$

Note that $G_i(x;\xi) = N\nabla_i F_i(x;\xi)$ for any $x$ and $\xi$, where $F_i(\cdot;\xi)$ is defined using the same way in (22) for $p = F(\cdot;\xi)$. We first highlight a frequently used inequality in the subsequent proof:

$$\|a+b\|^2 \quad\leqslant\quad 2\|a\|^2 + 2\|b\|^2. \tag{27}$$

We then start from the Lipschitzian property of $f(\cdot)$:

$$
\begin{aligned}
f(x_{k+1}) - f(x_k) \quad&\leqslant\quad \langle\nabla_{S_k}f(x_k), x_{k+1} - x_k\rangle + \frac{L_Y}{2}\|x_{k+1} - x_k\|^2 \\
&=\quad -\gamma_k\langle\nabla_{S_k}f(x_k), G_{S_k}(\hat{x}_k;\xi_k)\rangle \\
&\quad + \frac{1}{2}\gamma_k^2 L_Y\|G_{S_k}(\hat{x}_k;\xi_k)\|^2 \\
&\overset{(27)}{\leqslant}\quad -\gamma_k\langle\nabla_{S_k}f(x_k), G_{S_k}(\hat{x}_k;\xi_k)\rangle \\
&\quad + \gamma_k^2 L_Y\left\|\frac{N}{Y}\nabla_{S_k}F(\hat{x}_k;\xi_k)\right\|^2 \\
&\quad + \gamma_k^2 L_Y\left\|G_{S_k}(\hat{x}_k;\xi_k) - \frac{N}{Y}\nabla_{S_k}F(\hat{x}_k;\xi_k)\right\|^2. \tag{28}
\end{aligned}
$$

Next we consider the expectation of three items of the right hand side of (28). Let $\{i_k\}_{k=0}^{K}$ be independent random variables uniformly distributed over $\{1,\ldots,N\}$. Note that $i_k$ is just a dummy random variable independent of all $S_k$'s. Let $\Omega_k$ be the set consisting of all possible $S_k$'s. With this new variable, we can rewrite $\mathbb{E}_{S_k}\langle\nabla_{S_k}f(x_k), G_{S_k}(\hat{x}_k;\xi_k)\rangle$ by

$$
\begin{aligned}
&\mathbb{E}_{S_k}\langle\nabla_{S_k}f(x_k), G_{S_k}(\hat{x}_k;\xi_k)\rangle \\
&=\quad \frac{1}{Y}\mathbb{E}_{S_k}\sum_{m\in S_k}\langle\nabla_m f(x_k), G_m(\hat{x}_k;\xi_k)\rangle \\
&=\quad \frac{1}{Y|\Omega_k|}\sum_{S\in\Omega_k}\sum_{m\in S}\langle\nabla_m f(x_k), G_m(\hat{x}_k;\xi_k)\rangle \\
&=\quad \frac{1}{Y|\Omega_k|}\frac{|\Omega_k|Y}{N}\left\langle\nabla f(x_k), \sum_{n\in[N]}G_n(\hat{x}_k;\xi_k)\right\rangle \\
&=\quad \frac{1}{Y}Y\mathbb{E}_{i_k}\langle\nabla_{i_k}f(x_k), G_{i_k}(\hat{x}_k;\xi_k)\rangle \\
&=\quad \mathbb{E}_{i_k}\langle\nabla_{i_k}f(x_k), G_{i_k}(\hat{x}_k;\xi_k)\rangle, \tag{29}
\end{aligned}
$$

where the second last step is due to the fact that $S_k$ is selected uniformly randomly. Following similar derivation yields:

$$
\begin{aligned}
\mathbb{E}_{S_k}\left\|\frac{N}{Y}\nabla_{S_k}F(\hat{x}_k;\xi_k)\right\|^2 \quad&=\quad \frac{N^2}{Y^2}\mathbb{E}_{S_k}\sum_{m\in S_k}\|\nabla_m F(\hat{x}_k;\xi_k)\|^2 \\
&=\quad \frac{N^2}{Y^2}Y\mathbb{E}_{i_k}\|\nabla_{i_k}F(\hat{x}_k;\xi_k)\|^2
\end{aligned}
$$

$$
\begin{aligned}
&= \frac{N^2}{Y}\mathbb{E}_{i_k}\|\nabla_{i_k}F(\hat{x}_k;\xi_k)\|^2 \\
&= \frac{1}{Y}\mathbb{E}_{i_k}\|N\nabla_{i_k}F(\hat{x}_k;\xi_k)\|^2, \quad (30)
\end{aligned}
$$

and

$$
\begin{aligned}
&\mathbb{E}_{S_k}\left\|G_{S_k}(\hat{x}_k;\xi_k) - \frac{N}{Y}\nabla_{S_k}F(\hat{x}_k;\xi_k)\right\|^2 \\
&= \mathbb{E}_{S_k}\sum_{m\in S_k}\left\|\frac{1}{Y}G_m(\hat{x}_k;\xi_k) - \frac{N}{Y}\nabla_m F(\hat{x}_k;\xi_k)\right\|^2 \\
&= \frac{1}{Y^2}\mathbb{E}_{S_k}\sum_{m\in S_k}\|G_m(\hat{x}_k;\xi_k) - N\nabla_m F(\hat{x}_k;\xi_k)\|^2 \\
&= \frac{1}{Y^2}Y\mathbb{E}_{i_k}\|G_{i_k}(\hat{x}_k;\xi_k) - N\nabla_{i_k}F(\hat{x}_k;\xi_k)\|^2 \\
&= \frac{1}{Y}\mathbb{E}_{i_k}\|G_{i_k}(\hat{x}_k;\xi_k) - N\nabla_{i_k}F(\hat{x}_k;\xi_k)\|^2. \quad (31)
\end{aligned}
$$

We give two more equalities which will be used soon in the following:

$$
\begin{aligned}
\mathbb{E}_{\xi_k}\|\nabla F(\hat{x}_k;\xi_k)\|^2 &= \mathbb{E}_{\xi_k}(\|\nabla F(\hat{x}_k;\xi_k) - \nabla f(\hat{x}_k) + \nabla f(\hat{x}_k)\|^2) \\
&= \mathbb{E}_{\xi_k}(\|\nabla F(\hat{x}_k;\xi_k) - \nabla f(\hat{x}_k)\|^2 + \|\nabla f(\hat{x}_k)\|^2) \\
&\quad + 2\mathbb{E}_{\xi_k}\langle\nabla F(\hat{x}_k;\xi_k) - \nabla f(\hat{x}_k), \nabla f(\hat{x}_k)\rangle \\
&= \mathbb{E}_{\xi_k}(\|\nabla F(\hat{x}_k;\xi_k) - \nabla f(\hat{x}_k)\|^2 + \|\nabla f(\hat{x}_k)\|^2) \\
&\quad + 2\langle\nabla f(\hat{x}_k) - \nabla f(\hat{x}_k), \nabla f(\hat{x}_k)\rangle \\
&= \mathbb{E}_{\xi_k}(\|\nabla F(\hat{x}_k;\xi_k) - \nabla f(\hat{x}_k)\|^2 + \|\nabla f(\hat{x}_k)\|^2), \quad (32)
\end{aligned}
$$

and

$$
\begin{aligned}
&-2\langle\nabla_{i_k}f(x_k), \nabla_{i_k}f_{i_k}(\hat{x}_k)\rangle \\
&= \|\nabla_{i_k}f(x_k) - \nabla_{i_k}f_{i_k}(\hat{x}_k)\|^2 - \|\nabla_{i_k}f(x_k)\|^2 - \|\nabla_{i_k}f_{i_k}(\hat{x}_k)\|^2. \quad (33)
\end{aligned}
$$

We then put (29), (30), (31) into (28):

$$
\begin{aligned}
\mathbb{E}_{\xi_k,S_k}(f(x_{k+1}) - f(x_k)) &\leqslant -\gamma_k\mathbb{E}_{\xi_k,i_k}\langle\nabla_{i_k}f(x_k), G_{i_k}(\hat{x}_k;\xi_k)\rangle \\
&\quad + \gamma_k^2\frac{L_Y}{Y}\mathbb{E}_{\xi_k,i_k}\|N\nabla_{i_k}F(\hat{x}_k;\xi_k)\|^2 \\
&\quad + \gamma_k^2\frac{L_Y}{Y}\mathbb{E}_{\xi_k,i_k}\|G_{i_k}(\hat{x}_k;\xi_k) - N\nabla_{i_k}F(\hat{x}_k;\xi_k)\|^2 \\
&= -\gamma_k\mathbb{E}_{i_k}\langle\nabla_{i_k}f(x_k), N\nabla_{i_k}f_{i_k}(\hat{x}_k)\rangle \\
&\quad + \gamma_k^2\frac{L_Y}{Y}N\mathbb{E}_{\xi_k}\|\nabla F(\hat{x}_k;\xi_k)\|^2 \\
&\quad + \gamma_k^2\frac{L_Y}{Y}\mathbb{E}_{\xi_k,i_k}\|G_{i_k}(\hat{x}_k;\xi_k) - N\nabla_{i_k}F(\hat{x}_k;\xi_k)\|^2 \\
&\overset{(32)}{=} -\gamma_k N\mathbb{E}_{i_k}\langle\nabla_{i_k}f(x_k), \nabla_{i_k}f_{i_k}(\hat{x}_k)\rangle \\
&\quad + \gamma_k^2\frac{L_Y}{Y}N(\overbrace{\mathbb{E}_{\xi_k}\|\nabla F(\hat{x}_k;\xi_k) - \nabla f(\hat{x}_k)\|^2}^{\leqslant\sigma^2} + \|\nabla f(\hat{x}_k)\|^2) \\
&\quad + \gamma_k^2\frac{L_Y}{Y}\overbrace{\mathbb{E}_{\xi_k,i_k}\|G_{i_k}(\hat{x}_k;\xi_k) - N\nabla_{i_k}F(\hat{x}_k;\xi_k)\|^2}^{\leq N^2\omega/4 \text{ from (25)}} \\
&\overset{(33)}{\leqslant} -\frac{\gamma_k}{2}\left(\|\nabla f(x_k)\|^2 + N\mathbb{E}_{i_k}\|\nabla_{i_k}f_{i_k}(\hat{x}_k)\|^2\right) \\
&\quad + \frac{\gamma_k}{2}N\underbrace{\mathbb{E}_{i_k}\|\nabla_{i_k}f(x_k) - \nabla_{i_k}f_{i_k}(\hat{x}_k)\|^2}_{=:T_1}
\end{aligned}
$$

$$+\gamma_k^2 \frac{L_Y}{Y} N(\sigma^2 + \|\nabla f(\hat{x}_k)\|^2)$$

$$+\gamma_k^2 \frac{L_Y}{Y} N^2 \frac{\omega}{4}. \tag{34}$$

Then we study the upper bound for $T_1$:

$$
\begin{aligned}
T_1 &= \mathbb{E}_{i_k} \|\nabla_{i_k} f(x_k) - \nabla_{i_k} f_{i_k}(\hat{x}_k)\|^2 \\
&= \mathbb{E}_{i_k} \|\nabla_{i_k} f(x_k) - \nabla_{i_k} f(\hat{x}_k) + \nabla_{i_k} f(\hat{x}_k) - \nabla_{i_k} f_{i_k}(\hat{x}_k)\|^2 \\
&\overset{(27)}{\leqslant} 2\mathbb{E}_{i_k} \left( \|\nabla_{i_k} f(x_k) - \nabla_{i_k} f(\hat{x}_k)\|^2 + \|\nabla_{i_k} f(\hat{x}_k) - \nabla_{i_k} f_{i_k}(\hat{x}_k)\|^2 \right) \\
&\overset{(25)}{\leqslant} 2\mathbb{E}_{i_k} \|\nabla_{i_k} f(x_k) - \nabla_{i_k} f(\hat{x}_k)\|^2 + \frac{\omega}{2} \\
&\leqslant \frac{2}{N} L_T^2 \|x_k - \hat{x}_k\|^2 + \frac{\omega}{2} \\
&= \frac{2}{N} L_T^2 \underbrace{\left\| \sum_{j \in J(k)} (x_{j+1} - x_j) \right\|^2}_{=:T_2} + \frac{\omega}{2}. \tag{35}
\end{aligned}
$$

We then bound $T_2$ by

$$
\begin{aligned}
\mathbb{E}(T_2) &= \mathbb{E} \left\| \sum_{j \in J(k)} (x_{j+1} - x_j) \right\|^2 \\
&= \mathbb{E} \left\| \sum_{j \in J(k)} \gamma_j G_{S_j}(\hat{x}_j; \xi_j) \right\|^2 \\
&= \mathbb{E} \left\| \sum_{j \in J(k)} \left( \gamma_j \left( G_{S_j}(\hat{x}_j; \xi_j) - \frac{N}{Y} \sum_{m \in S_j} \nabla_m f_m(\hat{x}_j) \right) + \frac{N}{Y} \sum_{m \in S_j} \gamma_j \nabla_m f_m(\hat{x}_j) \right) \right\|^2 \\
&\overset{(27)}{\leqslant} 2\mathbb{E} \underbrace{\left\| \sum_{j \in J(k)} \gamma_j \left( G_{S_j}(\hat{x}_j; \xi_j) - \frac{N}{Y} \sum_{m \in S_j} \nabla_m f_m(\hat{x}_j) \right) \right\|^2}_{=:T_3} \\
&\quad + 2\frac{N^2}{Y^2} \mathbb{E} \underbrace{\left\| \sum_{j \in J(k)} \gamma_j \sum_{m \in S_j} \nabla_m f_m(\hat{x}_j) \right\|^2}_{=:T_4}. \tag{36}
\end{aligned}
$$

Next we show the upper bounds for $\mathbb{E}(T_4)$ and $\mathbb{E}(T_3)$ respectively:

$$
\begin{aligned}
&\mathbb{E}(T_4) \\
&= \mathbb{E} \left\| \sum_{j \in J(k)} \gamma_j \sum_{m \in S_j} \nabla_m f_m(\hat{x}_j) \right\|^2 \\
&= \mathbb{E} \left( \sum_{j \in J(k)} \gamma_j^2 \left\| \sum_{m \in S_j} \nabla_m f_m(\hat{x}_j) \right\|^2 \right) \\
&\quad + 2\mathbb{E} \left( \sum_{j,j' \in J(k), j > j'} \gamma_j \gamma_{j'} \left( \left\langle \sum_{m \in S_j} \nabla_m f_m(\hat{x}_j), \sum_{m' \in S_{j'}} \nabla_{m'} f_{m'}(\hat{x}_{j'}) \right\rangle \right) \right)
\end{aligned}
$$

$$
= \quad \mathbb{E}\left( \sum_{j \in J(k)} \gamma_j^2 Y \mathbb{E}_{i_j} \left\| \nabla_{i_j} f_{i_j}(\hat{x}_j) \right\|^2 \right)
$$

$$
+ 2\mathbb{E}\left( \sum_{j,j' \in J(k), j>j'} \gamma_j \gamma_{j'} \left\langle \mathbb{E}_{S_j} \sum_{m \in S_j} \nabla_m f_m(\hat{x}_j), \sum_{m' \in S_{j'}} \nabla_{m'} f_{m'}(\hat{x}_{j'}) \right\rangle \right)
$$

$$
\leqslant \quad \sum_{j \in J(k)} \gamma_j^2 Y \mathbb{E} \left\| \nabla_{i_j} f_{i_j}(\hat{x}_j) \right\|^2
$$

$$
+ \mathbb{E}\left( \sum_{j,j' \in J(k), j>j'} \gamma_j \gamma_{j'} \left( \frac{1}{\alpha} \left\| \mathbb{E}_{S_j} \sum_{m \in S_j} \nabla_m f_m(\hat{x}_j) \right\|^2 + \alpha \mathbb{E}_{S_{j'}} \left\| \sum_{m' \in S_{j'}} \nabla_{m'} f_{m'}(\hat{x}_{j'}) \right\|^2 \right) \right)
$$

$$
\leqslant \quad \sum_{j \in J(k)} \gamma_j^2 Y \mathbb{E} \left\| \nabla_{i_j} f_{i_j}(\hat{x}_j) \right\|^2
$$

$$
+ \mathbb{E}\left( \sum_{j,j' \in J(k), j>j'} \gamma_j \gamma_{j'} \left( \frac{1}{\alpha} \left\| Y \mathbb{E}_{i_j} \nabla_{i_j} f_{i_j}(\hat{x}_j) \right\|^2 + \alpha \mathbb{E}_{S_{j'}} \left\| \sum_{m' \in S_{j'}} \nabla_{m'} f_{m'}(\hat{x}_{j'}) \right\|^2 \right) \right)
$$

$$
\leqslant \quad \sum_{j \in J(k)} \gamma_j^2 Y \mathbb{E} \left\| \nabla_{i_j} f_{i_j}(\hat{x}_j) \right\|^2
$$

$$
+ \mathbb{E}\left( \sum_{j,j' \in J(k), j>j'} \gamma_j \gamma_{j'} \left( \frac{Y^2}{\alpha N} \mathbb{E}_{i_j} \left\| \nabla_{i_j} f_{i_j}(\hat{x}_j) \right\|^2 + \alpha Y \mathbb{E}_{i_{j'}} \left\| \nabla_{i_{j'}} f_{\mu_{i_{j'}},i_{j'}}(\hat{x}_{j'}) \right\|^2 \right) \right)
$$

$$
= \quad \sum_{j \in J(k)} \gamma_j \left( \gamma_j Y + \alpha Y \sum_{j'>j,j' \in J(k)} \gamma_{j'} + \frac{Y^2 \sum_{j'<j,j' \in J(k)} \gamma_{j'}}{\alpha N} \right) \mathbb{E} \left\| \nabla_{i_j} f_{i_j}(\hat{x}_j) \right\|^2
$$

$$
= \quad \sum_{j \in J(k)} \gamma_j \left( \gamma_j Y + \frac{Y^{3/2} \sum_{j' \in J(k) \setminus \{j\}} \gamma_{j'}}{\sqrt{N}} \right) \mathbb{E} \left\| \nabla_{i_j} f_{i_j}(\hat{x}_j) \right\|^2 ,
$$

where we set $\alpha = \sqrt{Y/N}$ in the last step. We next consider $\mathbb{E}(T_3)$

$$
\mathbb{E}(T_3) \quad = \quad \mathbb{E}\left( \left\| \sum_{j \in J(k)} \gamma_j \left( G_{S_j}(\hat{x}_j; \xi_j) - \frac{N}{Y} \sum_{m \in S_j} \nabla_m f_m(\hat{x}_j) \right) \right\|^2 \right)
$$

$$
= \quad \mathbb{E}\left( \sum_{j \in J(k)} \gamma_j^2 \left\| G_{S_j}(\hat{x}_j; \xi_j) - \frac{N}{Y} \sum_{m \in S_j} \nabla_m f_m(\hat{x}_j) \right\|^2 \right)
$$

$$
= \quad \mathbb{E}\left( \sum_{j \in J(k)} \frac{\gamma_j^2}{Y^2} \left\| Y G_{S_j}(\hat{x}_j; \xi_j) - N \sum_{m \in S_j} \nabla_m f_m(\hat{x}_j) \right\|^2 \right)
$$

$$
= \quad \mathbb{E}\left( \sum_{j \in J(k)} \frac{\gamma_j^2}{Y} \left\| G_{i_j}(\hat{x}_j; \xi_j) - N \nabla_{i_j} f_{i_j}(\hat{x}_j) \right\|^2 \right), \tag{37}
$$

where the last equality is due to the same reason of (31) and the second last equality is from

$$
2\mathbb{E}\left( \sum_{j>j';j,j' \in J(k)} \gamma_j \gamma_{j'} \langle \Gamma_j, \Gamma_{j'} \rangle \right)
$$

$$= 2\mathbb{E}\left(\sum_{j>j';j,j'\in J(k)} \gamma_j\gamma_{j'} \left\langle \mathbb{E}_{\xi_j} G_{S_j}(\hat{x}_j;\xi_j) - \frac{N}{Y}\sum_{m\in S_j} \nabla_m f_m(\hat{x}_j), \Gamma_{j'} \right\rangle\right)$$

$$= 0,$$

where

$$\Gamma_j := G_{S_j}(\hat{x}_j;\xi_j) - \frac{N}{Y}\sum_{m\in S_j} \nabla_m f_m(\hat{x}_j)$$

$$\Gamma_{j'} := G_{S_{j'}}(\hat{x}_{j'};\xi_{j'}) - \frac{N}{Y}\sum_{m'\in S_{j'}} \nabla_{m'} f_{m'}(\hat{x}_{j'}).$$

Note that we have the following inequality

$$\mathbb{E}_{\xi,i}\|G_i(x;\xi) - N\nabla_i f_i(x)\|^2$$

$$= \mathbb{E}_{\xi,i}\|G_i(x;\xi) - N\nabla_i F(x;\xi) + N\nabla_i F(x;\xi) - N\nabla_i f(x) + N\nabla_i f(x) - N\nabla_i f_i(x)\|^2$$

$$\leqslant 3\mathbb{E}_{\xi,i}(\|G_i(x;\xi) - N\nabla_i F(x;\xi)\|^2 + N^2\|\nabla_i F(x;\xi) - \nabla_i f(x)\|^2 + N^2\|\nabla_i f(x) - \nabla_i f_i(x)\|^2)$$

$$\leqslant 3\mathbb{E}_\xi\left(\frac{N^2\omega}{2} + N\|\nabla F(x;\xi) - \nabla f(x)\|^2\right)$$

$$\overset{(25)}{\leqslant} \frac{3N^2\omega}{2} + 3N\sigma^2,$$

where the first step uses $\|a+b+c\|^2 \leqslant 3(\|a\|^2 + \|b\|^2 + \|c\|^2)$. Applying it into (37) yields:

$$\mathbb{E}(T_3) \leq \left(\frac{3N^2\omega}{2Y} + \frac{3N\sigma^2}{Y}\right)\sum_{j\in J(k)} \gamma_j^2,$$

Applying the upper bounds of $\mathbb{E}(T_3)$ and $\mathbb{E}(T_4)$ to (36) we obtain

$$\mathbb{E}(T_2) \leqslant 2\mathbb{E}(T_3) + 2\frac{N^2}{Y^2}\mathbb{E}(T_4)$$

$$\leqslant 2\left(\frac{3N^2\omega}{2Y} + \frac{3N\sigma^2}{Y}\right)\sum_{j\in J(k)} \gamma_j^2$$

$$+ 2\frac{N^2}{Y^2}\sum_{j\in J(k)} \gamma_j\left(\gamma_j Y + \frac{Y^{3/2}\sum_{j'\in J(k)\setminus\{j\}}\gamma_{j'}}{\sqrt{N}}\right)\mathbb{E}\|\nabla_{i_j} f_{i_j}(\hat{x}_j)\|^2.$$

We apply the upper bound of $\mathbb{E}(T_2)$ to (35):

$$\mathbb{E}(T_1) \leqslant \frac{2}{N}L_T^2\mathbb{E}(T_2) + \frac{\omega}{2}$$

$$\leqslant \frac{4L_T^2}{Y}\left(\frac{3N\omega}{2} + 3\sigma^2\right)\sum_{j\in J(k)}\gamma_j^2 + \frac{\omega}{2}$$

$$+ 4L_T^2\frac{N}{Y^2}\sum_{j\in J(k)}\gamma_j\left(\gamma_j Y + \frac{Y^{3/2}\sum_{j'\in J(k)\setminus\{j\}}\gamma_{j'}}{\sqrt{N}}\right)\mathbb{E}\left\|\nabla_{i_j} f_{i_j}(\hat{x}_j)\right\|^2.$$

Substitute the upper bound of $\mathbb{E}(T_1)$ into (34)

$$\mathbb{E}(f(x_{k+1}) - f(x_k))$$

$$\leqslant -\frac{\gamma_k}{2}\left(\mathbb{E}\|\nabla f(x_k)\|^2 + N\mathbb{E}|\nabla_{i_k} f_{i_k}(\hat{x}_k)|^2\right)$$

$$+ \frac{\gamma_k}{2}N\mathbb{E}(T_1) + \gamma_k^2\frac{L_Y}{Y}N(\sigma^2 + \mathbb{E}\|\nabla f(\hat{x}_k)\|^2) + \gamma_k^2\frac{L_Y}{Y}N^2\frac{\omega}{4}$$

$$\leqslant -\frac{\gamma_k}{2}\left(\mathbb{E}\|\nabla f(x_k)\|^2 + N\mathbb{E}\|\nabla_{i_k} f_{i_k}(\hat{x}_k)\|^2\right)$$

$$+ \frac{\gamma_k}{2} N \left( 4 L_T^2 \frac{\left( \frac{3N\omega}{2} + 3\sigma^2 \right)}{Y} \sum_{j \in J(k)} \gamma_j^2 + \frac{\omega}{2} \right)$$

$$+ 2\gamma_k \left( L_T^2 \frac{N^2}{Y^2} \sum_{j \in J(k)} \gamma_j \left( \gamma_j Y + \frac{Y^{3/2} \sum_{j' \in J(k) \setminus \{j\}} \gamma_{j'}}{\sqrt{N}} \right) \mathbb{E} \left\| \nabla_{i_j} f_{i_j}(\hat{x}_j) \right\|^2 \right)$$

$$+ \gamma_k^2 \frac{L_Y}{Y} N (\sigma^2 + \mathbb{E} \| \nabla f(\hat{x}_k) \|^2) + \gamma_k^2 \frac{L_Y}{Y} N^2 \frac{\omega}{4}$$

$$\overset{\text{rearrange}}{=} \quad -\frac{\gamma_k}{2} \mathbb{E} \| \nabla f(x_k) \|^2 + \gamma_k^2 \frac{L_Y}{Y} N^2 \underbrace{\frac{1}{N} \mathbb{E} \| \nabla f(\hat{x}_k) \|^2}_{=: T_5}$$

$$- \left( \frac{\gamma_k N}{2} \right) \mathbb{E} \left\| \nabla_{i_k} f_{i_k}(\hat{x}_k) \right\|^2$$

$$+ 2\gamma_k \left( L_T^2 \frac{N^2}{Y^2} \sum_{j \in J(k)} \gamma_j \left( \gamma_j Y + \frac{Y^{3/2} \sum_{j' \in J(k) \setminus \{j\}} \gamma_{j'}}{\sqrt{N}} \right) \mathbb{E} \left\| \nabla_{i_j} f_{i_j}(\hat{x}_j) \right\|^2 \right)$$

$$+ \frac{\gamma_k}{2} N \left( \frac{4 L_T^2}{Y} \left( \frac{3N\omega}{2} + 3\sigma^2 \right) \sum_{j \in J(k)} \gamma_j^2 + \frac{\omega}{2} \right)$$

$$+ \gamma_k^2 \frac{L_Y}{Y} N \sigma^2 + \gamma_k^2 \frac{L_Y}{Y} N^2 \frac{\omega}{4}. \tag{38}$$

Note that

$$
\begin{aligned}
T_5 &= \frac{1}{N} \mathbb{E} \| \nabla f(\hat{x}_k) \|^2 \\
&= \mathbb{E} \| \nabla_{i_k} f(\hat{x}_k) \|^2 \\
&\overset{(27)}{\leqslant} 2 \mathbb{E} \| \nabla_{i_k} f_{i_k}(\hat{x}_k) \|^2 + 2 \mathbb{E} \| \nabla_{i_k} f(\hat{x}_k) - \nabla_{i_k} f_{i_k}(\hat{x}_k) \|^2 \\
&\overset{(25)}{\leqslant} 2 \mathbb{E} \| \nabla_{i_k} f_{i_k}(\hat{x}_k) \|^2 + \frac{\omega}{2}.
\end{aligned}
$$

Substitute this upper bound of $T_5$ into (38):

$$\mathbb{E}(f(x_{k+1}) - f(x_k))$$

$$\leqslant \quad -\frac{\gamma_k}{2} \mathbb{E} \| \nabla f(x_k) \|^2 + \gamma_k^2 \frac{L_Y}{2Y} N^2 \omega$$

$$- \left( \frac{\gamma_k N}{2} - 2\gamma_k^2 \frac{L_Y}{Y} N^2 \right) \mathbb{E} \left\| \nabla_{i_k} f_{i_k}(\hat{x}_k) \right\|^2$$

$$+ 2\gamma_k \left( L_T^2 \frac{N^2}{Y^2} \sum_{j \in J(k)} \gamma_j \left( \gamma_j Y + \frac{Y^{3/2} \sum_{j' \in J(k) \setminus \{j\}} \gamma_{j'}}{\sqrt{N}} \right) \mathbb{E} \left\| \nabla_{i_j} f_{i_j}(\hat{x}_j) \right\|^2 \right)$$

$$+ \frac{\gamma_k}{2} N \left( \frac{4 L_T^2}{Y} \left( \frac{3N\omega}{2} + 3\sigma^2 \right) \sum_{j \in J(k)} \gamma_j^2 + \frac{\omega}{2} \right)$$

$$+ \gamma_k^2 \frac{L_Y}{Y} N \sigma^2 + \gamma_k^2 \frac{L_Y}{Y} N^2 \frac{\omega}{4}$$

$$\leqslant \quad -\frac{\gamma_k}{2} \mathbb{E} \| \nabla f(x_k) \|^2 - \left( \frac{\gamma_k N}{2} - 2\gamma_k^2 \frac{L_Y}{Y} N^2 \right) \mathbb{E} \left\| \nabla_{i_k} f_{i_k}(\hat{x}_k) \right\|^2$$

$$+ 2\gamma_k \left( L_T^2 \frac{N^2}{Y^2} \sum_{j \in J(k)} \gamma_j \left( \gamma_j Y + \frac{Y^{3/2} \sum_{j' \in J(k) \setminus \{j\}} \gamma_{j'}}{\sqrt{N}} \right) \mathbb{E} \left\| \nabla_{i_j} f_{i_j}(\hat{x}_j) \right\|^2 \right)$$

$$+ 2\gamma_k N \frac{L_T^2}{Y} \left( \frac{3N\omega}{2} + 3\sigma^2 \right) \sum_{j \in J(k)} \gamma_j^2$$

$$+\gamma_k^2 \frac{L_Y}{Y} N\sigma^2 + \gamma_k^2 \frac{L_Y}{Y} N^2\omega + \frac{\gamma_k}{4} N\omega. \tag{39}$$

Summarizing (39) from $k = 0$ to $k = K$ (note that $\Theta_k$ is defined in (26)) yields:

$$\frac{1}{2} \sum_{k=0}^{K} \gamma_k \mathbb{E}\|\nabla f(x_k)\|^2$$

$$\leqslant \quad f(x_0) - f(x_{K+1}) - \sum_{k=0}^{K} \Theta_k \mathbb{E} \|\nabla_{i_k} f_{i_k}(\hat{x}_k)\|^2$$

$$+ 2N\frac{L_T^2}{Y}\left(\frac{3N\omega}{2} + 3\sigma^2\right)\sum_{k=0}^{K}\left(\gamma_k \sum_{j \in J(k)} \gamma_j^2\right)$$

$$+ \left(\frac{L_Y}{Y}N\sigma^2 + \frac{L_Y}{Y}N^2\omega\right)\sum_{k=0}^{K}\gamma_k^2 + \frac{1}{4}N\omega\sum_{k=0}^{K}\gamma_k$$

$$\overset{\Theta_k \geq 0}{\leqslant} \quad f(x_0) - f^* + 2N\frac{L_T^2}{Y}\left(\frac{3N\omega}{2} + 3\sigma^2\right)\sum_{k=0}^{K}\left(\gamma_k \sum_{j \in J(k)} \gamma_j^2\right)$$

$$+ \left(\frac{L_Y}{Y}N\sigma^2 + \frac{L_Y}{Y}N^2\omega\right)\sum_{k=0}^{K}\gamma_k^2 + \frac{1}{4}N\omega\sum_{k=0}^{K}\gamma_k.$$

It completes the proof. □

**Corollary 8.** *Set all steplength $\gamma_k$ to be a constant $\gamma$ in Algorithm 1*

$$\gamma = \frac{Y}{N}\frac{1}{2L_Y\chi}, \tag{40}$$

*where $\chi$ satisfies*

$$\chi \geq \sqrt{1 + \frac{L_T^2}{L_Y^2}\left(\frac{Y}{N} + \frac{Y^{3/2}T}{N^{3/2}}\right)T + 1}. \tag{41}$$

*It ensures the following convergence rate*

$$\frac{1}{2K}\sum_{k=0}^{K}\mathbb{E}\|\nabla f(x_k)\|^2 \quad \leq \quad \frac{2(f(x_0) - f^*)L_Y N}{KY}\chi + \frac{N\omega + \sigma^2}{\chi}\left(1 + \frac{2L_T^2 YT}{L_Y^2 N}\frac{1}{\chi}\right) + N\omega.$$

*Proof.* To apply Theorem 7, we first verify that the choice of $\gamma$ in (40) satisfies the prerequisite (26). Letting $\gamma_k = \gamma$, the prerequisite (26) in Theorem 7 reduces to

$$\frac{N\gamma}{2} - 2\gamma^2 \frac{L_Y}{Y}N^2 - 2L_T^2\frac{N^2}{Y^2}\gamma\left(\gamma Y + \frac{Y^{3/2}T\gamma}{\sqrt{N}}\right)T\gamma \quad \geqslant \quad 0, \forall k,$$

or equivalently

$$\frac{L_T^2}{Y^2}\gamma^2 \underbrace{\left(Y + \frac{Y^{3/2}T}{\sqrt{N}}\right)T}_{=:l} + \gamma\frac{L_Y}{Y} - \frac{1}{4N} \quad \leqslant \quad 0, \forall k.$$

Here we denote $\left(Y + \frac{Y^{3/2}T}{\sqrt{N}}\right)T$ by $l$ for short here.

To satisfy the above inequality, it suffices to show

$$\gamma \quad \leqslant \quad \frac{-L_Y/Y + \sqrt{L_Y^2/Y^2 + (L_T^2/Y^2)l/N}}{2L_T^2 l/Y^2}$$

$$= \quad Y\frac{-L_Y + \sqrt{L_Y^2 + L_T^2 l/N}}{2L_T^2 l}$$

$$
\begin{aligned}
&= \ Y\frac{L_Y}{2L_T^2 l}\left(\sqrt{1+\frac{L_T^2 l}{L_Y^2 N}}-1\right)\\
&= \ Y\frac{1}{2L_Y N}\frac{\sqrt{1+\chi'}-1}{\chi'}\\
&= \ Y\frac{1}{2L_Y N}\frac{1}{\chi_0},
\end{aligned}
$$

where

$$
\chi' \ := \ \frac{L_T^2 l}{L_Y^2 N} = \frac{L_T^2\left(Y+\frac{Y^{3/2}T}{\sqrt{N}}\right)T}{L_Y^2 N},
$$

$$
\chi_0 \ := \ \sqrt{1+\chi'}+1 = \sqrt{1+\frac{L_T^2\left(Y+\frac{Y^{3/2}T}{\sqrt{N}}\right)T}{L_Y^2 N}}+1.
$$

Due to $\chi \geq \chi_0$, the choice of $\gamma$ in (26) satisfies the prerequisite (26).

Now by applying Theorem 7

$$
\begin{aligned}
\frac{1}{2}\sum_{k=0}^{K}\gamma_k\mathbb{E}\|\nabla f(x_k)\|^2 \ \leqslant \ & f(x_0)-f^* + 2N\frac{L_T^2}{Y}\left(\frac{3N\omega}{2}+3\sigma^2\right)\sum_{k=0}^{K}\left(\gamma_k\sum_{j\in J(k)}\gamma_j^2\right)\\
&+\left(\frac{L_Y}{Y}N\sigma^2+\frac{L_Y}{Y}N^2\omega\right)\sum_{k=0}^{K}\gamma_k^2+\frac{1}{4}N\omega\sum_{k=0}^{K}\gamma_k.
\end{aligned}
$$

and letting $\gamma_k=\gamma$ and dividing both sides by $K\gamma$, we obtain

$$
\begin{aligned}
\frac{1}{2K}\sum_{k=0}^{K}\mathbb{E}\|\nabla f(x_k)\|^2 \ \leqslant \ & \frac{f(x_0)-f^*}{K\gamma}+2N\frac{L_T^2}{Y}\left(\frac{3N\omega}{2}+3\sigma^2\right)T\gamma^2\\
&+\left(\frac{L_Y}{Y}N\sigma^2+\frac{L_Y}{Y}N^2\omega\right)\gamma+\frac{1}{4}N\omega. \qquad (42)
\end{aligned}
$$

Substituting $\gamma$ into (42), we have

$$
\begin{aligned}
\frac{1}{2K}\sum_{k=0}^{K}\mathbb{E}\|\nabla f(x_k)\|^2 \ \leqslant \ & \frac{f(x_0)-f^*}{K\gamma}+2N\frac{L_T^2}{Y}\left(\frac{3N\omega}{2}+3\sigma^2\right)T\gamma^2\\
&+\left(\frac{L_Y}{Y}N\sigma^2+\frac{L_Y}{Y}N^2\omega\right)\gamma+\frac{1}{4}N\omega\\
= \ & \frac{2(f(x_0)-f^*)L_Y N}{KY}\chi\\
&+\frac{L_T^2 YT}{2L_Y^2 N}\left(\frac{3N\omega}{2}+3\sigma^2\right)\frac{1}{\chi^2}+(\sigma^2+N\omega)\frac{1}{2\chi}+\frac{1}{4}N\omega\\
\leqslant \ & \frac{2(f(x_0)-f^*)L_Y N}{KY}\chi\\
&+\frac{2L_T^2 YT}{L_Y^2 N}(N\omega+\sigma^2)\frac{1}{\chi^2}+(\sigma^2+N\omega)\frac{1}{\chi}+N\omega\\
= \ & \frac{2(f(x_0)-f^*)L_Y N}{KY}\chi+\frac{N\omega+\sigma^2}{\chi}\left(1+\frac{2L_T^2 YT}{L_Y^2 N}\frac{1}{\chi}\right)+N\omega,
\end{aligned}
$$

which completing the proof. $\qquad\square$

**Theorem 9.** *Set all steplength $\gamma_k$ to be a constant $\gamma$ in Algorithm 1*

$$
\gamma = \frac{Y}{N}\frac{1}{2L_Y\chi},
$$

*where*

$$\chi = \sqrt{\frac{\alpha_1^2}{K(N\omega + \sigma^2)\alpha_2 + \alpha_1}} + \sqrt{K(N\omega + \sigma^2)\alpha_2}. \tag{43}$$

*It ensures the following convergence rate*

$$\frac{\sum_{k=0}^{K} \mathbb{E}\|\nabla f(x_k)\|^2}{2K} \leqslant \frac{2\alpha_1}{K\alpha_2\sqrt{K(N\omega + \sigma^2)\alpha_2 + \alpha_1}} + \frac{3\sqrt{N\omega + \sigma^2}}{\sqrt{K\alpha_2}} + \frac{2L_T^2 YT}{L_Y^2 NK\alpha_2} + N\omega.$$

*Proof.* In order to apply Corollary 8, we first verify that the choice of $\chi$ in (43) satisfies the requirement in (41). (43) suggests that

$$\chi^2 \geqslant \frac{\alpha_1^2}{K(N\omega + \sigma^2)\alpha_2 + \alpha_1} + K(N\omega + \sigma^2)\alpha_2 \geq \alpha_1.$$

where the second inequality is due to that $K = 0$ minimize the second part. Also note that

$$\begin{aligned}
\alpha_1 &= 4\left(1 + \frac{L_T^2\left(Y + \frac{Y^{3/2}T}{\sqrt{N}}\right)T}{L_Y^2 N}\right) \\
&= \left(2\sqrt{1 + \frac{L_T^2}{L_Y^2}\left(\frac{Y}{N} + \frac{Y^{3/2}T}{N^{3/2}}\right)T}\right)^2 \\
&\geqslant \left(\sqrt{1 + \frac{L_T^2}{L_Y^2}\left(\frac{Y}{N} + \frac{Y^{3/2}T}{N^{3/2}}\right)T} + 1\right)^2.
\end{aligned}$$

Therefore the choice of $\chi$ in (43) satisfies the condition in (41) required by Corollary 8. Applying Corollary 8 yields the following convergence rate

$$\begin{aligned}
&\frac{1}{2K}\sum_{k=0}^{K}\mathbb{E}\|\nabla f(x_k)\|^2 \\
\leqslant\ & \frac{2(f(x_0) - f^*)L_Y N}{KY}\chi + \frac{N\omega + \sigma^2}{\chi}\left(1 + \frac{2L_T^2 YT}{L_Y^2 N}\frac{1}{\chi}\right) + N\omega \\
\overset{(43)}{=}\ & \frac{2(f(x_0) - f^*)L_Y N}{KY}\sqrt{\frac{\alpha_1^2}{K(N\omega + \sigma^2)\alpha_2 + \alpha_1}} \\
& + \frac{2(f(x_0) - f^*)L_Y N}{\sqrt{K}Y}\sqrt{(N\omega + \sigma^2)\alpha_2} \\
& + \underbrace{\frac{N\omega + \sigma^2}{\sqrt{\frac{\alpha_1^2}{K(N\omega + \sigma^2)\alpha_2 + \alpha_1}} + \sqrt{K(N\omega + \sigma^2)\alpha_2}}}_{\text{discard}} \\
& + \frac{2L_T^2 YT}{L_Y^2 N}\underbrace{\frac{N\omega + \sigma^2}{\left(\underbrace{\sqrt{\frac{\alpha_1^2}{K(N\omega + \sigma^2)\alpha_2 + \alpha_1}}}_{\text{discard}} + \sqrt{K(N\omega + \sigma^2)\alpha_2}\right)^2}} \\
& + N\omega \\
\leqslant\ & \frac{2(f(x_0) - f^*)L_Y N\alpha_1}{KY\sqrt{K(N\omega + \sigma^2)\alpha_2 + \alpha_1}} \\
& + \frac{2(f(x_0) - f^*)L_Y N\sqrt{(N\omega + \sigma^2)\alpha_2}}{\sqrt{K}Y}
\end{aligned}$$

$$+\frac{\sqrt{N\omega+\sigma^2}}{\sqrt{K\alpha_2}}+\frac{2L_T^2YT}{L_Y^2NK\alpha_2}+N\omega$$

$$=\frac{2\alpha_1}{K\alpha_2\sqrt{K(N\omega+\sigma^2)\alpha_2+\alpha_1}}+\frac{3\sqrt{N\omega+\sigma^2}}{\sqrt{K\alpha_2}}+\frac{2L_T^2YT}{L_Y^2NK\alpha_2}+N\omega.$$

It completes the proof. □

**Proof to Theorem 1**

*Proof.* Note that in Theorem 1 we have the same steplength as in Theorem 9, so we can safely apply Theorem 9 to obtain

$$\frac{\sum_{k=0}^K\mathbb{E}\|\nabla f(x_k)\|^2}{2K}$$

$$\leqslant\frac{2\alpha_1}{K\alpha_2\sqrt{K(N\omega+\sigma^2)\alpha_2+\alpha_1}}+\frac{3\sqrt{N\omega+\sigma^2}}{\sqrt{K\alpha_2}}+\frac{2L_T^2YT}{L_Y^2NK\alpha_2}+N\omega$$

$$\overset{(7)}{=}\frac{8\left(1+\frac{L_T^2T}{L_Y^2N}\left(Y+\frac{Y^{3/2}T}{\sqrt{N}}\right)\right)}{K\alpha_2\sqrt{K(N\omega+\sigma^2)\alpha_2+4+\underbrace{\frac{4L_T^2T}{L_Y^2N}\left(Y+\frac{Y^{3/2}T}{\sqrt{N}}\right)}_{\text{discard}}}}$$

$$+\frac{3\sqrt{N\omega+\sigma^2}}{\sqrt{K\alpha_2}}+\frac{2L_T^2YT}{L_Y^2NK\alpha_2}+N\omega$$

$$\leqslant\frac{8}{K\alpha_2\sqrt{K(N\omega+\sigma^2)\alpha_2+4}}+\frac{8\frac{L_T^2T}{L_Y^2N}\left(Y+\frac{Y^{3/2}T}{\sqrt{N}}\right)}{K\alpha_2\sqrt{K(N\omega+\sigma^2)\alpha_2+4}}$$

$$+\frac{3\sqrt{N\omega+\sigma^2}}{\sqrt{K\alpha_2}}+\frac{2L_T^2YT}{L_Y^2NK\alpha_2}+N\omega. \tag{44}$$

Next from the condition of $T$ in Theorem 1 and the definition of $\alpha_3$, we can obtain

$$\frac{L_T^2\left(Y+\frac{Y^{3/2}T}{\sqrt{N}}\right)T}{L_Y^2N}\leqslant K(N\omega+\sigma^2)\alpha_2+4=\alpha_3\frac{L_T^2}{L_Y^2}. \tag{45}$$

To see why it is true, it suffices to show that

$$T \leqslant \frac{-Y\sqrt{N}+\sqrt{NY^2+4Y^{3/2}N^{3/2}\alpha_3}}{2Y^{3/2}}$$

$$= \frac{-\sqrt{N}+\sqrt{N+4Y^{-1/2}N^{3/2}\alpha_3}}{2Y^{1/2}}$$

$$= \frac{\sqrt{N}}{2Y^{1/2}}\left(\sqrt{1+4Y^{-1/2}N^{1/2}\alpha_3}-1\right), \tag{46}$$

which is implied by the prerequisite for $T$ in Theorem 1.

Then we apply (45) to (44) and obtain

$$\frac{\sum_{k=0}^K\mathbb{E}\|\nabla f(x_k)\|^2}{2K}$$

$$\leqslant\frac{8}{K\alpha_2\sqrt{K(N\omega+\sigma^2)\alpha_2+4}}+\frac{8\frac{L_T^2\left(Y+\frac{Y^{3/2}T}{\sqrt{N}}\right)T}{L_Y^2N}}{K\alpha_2\sqrt{K(N\omega+\sigma^2)\alpha_2+4}}$$

$$+\frac{2L_T^2YT}{L_Y^2NK\alpha_2}+\frac{3\sqrt{N\omega+\sigma^2}}{\sqrt{K\alpha_2}}+N\omega$$

$$\overset{(46)}{\leqslant} \quad \frac{8}{K\alpha_2\sqrt{K(N\omega+\sigma^2)\alpha_2+4}} + \frac{8\sqrt{K(N\omega+\sigma^2)\alpha_2+4}}{K\alpha_2}$$

$$+\frac{L_T^2\sqrt{NY}\left(\sqrt{1+4Y^{-1/2}N^{1/2}\alpha_3}-1\right)}{L_Y^2 NK\alpha_2} + \frac{3\sqrt{N\omega+\sigma^2}}{\sqrt{K\alpha_2}} + N\omega$$

$$\leqslant \quad \frac{8}{K\alpha_2\sqrt{K(N\omega+\sigma^2)\alpha_2+4}} + \frac{8\sqrt{K(N\omega+\sigma^2)\alpha_2}}{K\alpha_2} + \frac{16}{K\alpha_2} + \frac{3\sqrt{N\omega+\sigma^2}}{\sqrt{K\alpha_2}} + N\omega$$

$$+\frac{1}{K\alpha_2}\frac{L_T^2}{L_Y^2}\frac{\sqrt{Y}\left(\sqrt{1+4Y^{-1/2}N^{1/2}\alpha_3}-1\right)}{\sqrt{N}}$$

$$= \quad \frac{1}{K\alpha_2}\left(16 + \frac{L_T^2}{L_Y^2}\frac{\sqrt{Y}\left(\sqrt{1+4Y^{-1/2}N^{1/2}\alpha_3}-1\right)}{\sqrt{N}} + \overbrace{\frac{8}{\sqrt{K(N\omega+\sigma^2)\alpha_2+4}}}^{\leq 4}\right)$$

$$+\frac{11\sqrt{N\omega+\sigma^2}}{\sqrt{K\alpha_2}} + N\omega$$

$$\leqslant \quad \frac{1}{K\alpha_2}\left(20 + \frac{L_T^2}{L_Y^2}\frac{\sqrt{Y}\left(\sqrt{1+4Y^{-1/2}N^{1/2}\alpha_3}-1\right)}{\sqrt{N}}\right) + \frac{11\sqrt{N\omega+\sigma^2}}{\sqrt{K\alpha_2}} + N\omega$$

$$= \quad \frac{20}{K\alpha_2} + \frac{1}{K\alpha_2}\left(\frac{L_T^2}{L_Y^2}\frac{\sqrt{Y}\left(\sqrt{1+4Y^{-1/2}N^{1/2}\alpha_3}-1\right)}{\sqrt{N}} + 11\sqrt{N\omega+\sigma^2}\sqrt{K\alpha_2}\right) + N\omega.$$

It completes the proof. □

### .3 Proofs to Corollaries

We prove all corollaries using Theorem 1 in this subsection.

**Proof to Corollary 2**

*Proof.* For ASCD, letting $\sigma = 0$, $\omega = 0$, and $Y = 1$ in Thereon 1, we have

$$\gamma^{-1} = 2L_{\max}N\left(\sqrt{\frac{\alpha_1^2}{K(N\omega+\sigma^2)\alpha_2+\alpha_1}} + \sqrt{K(N\omega+\sigma^2)\alpha_2}\right)$$

$$= 2L_{\max}N\sqrt{\alpha_1},$$

and the prerequisite becomes

$$T \leqslant \frac{\sqrt{N}}{2}\left(\sqrt{1+4\alpha_3 N^{1/2}}-1\right)$$

$$= \frac{\sqrt{N}}{2}\left(\sqrt{1+4\frac{L_Y^2}{L_T^2}(K(N\omega+\sigma^2)\alpha_2+4)N^{1/2}}-1\right)$$

$$= \frac{\sqrt{N}}{2}\left(\sqrt{1+16\frac{L_{\max}^2}{L_T^2}N^{1/2}}-1\right).$$

$$= O(N^{3/4})$$

The convergence rate turns out to be

$$\frac{\sum_{k=0}^{K}\mathbb{E}\|\nabla f(x_k)\|^2}{2K}$$

$$\leqslant \frac{1}{K\alpha_2}\left(20 + \frac{L_T^2}{L_{\max}^2}\frac{\sqrt{1+16N^{1/2}\frac{L_{\max}^2}{L_T^2}}-1}{\sqrt{N}}\right)$$

$$= \frac{(f(x_0) - f^*)L_{\max}N}{K}\left(20 + \frac{L_T^2}{L_{\max}^2}\frac{\sqrt{1 + 16N^{1/2}\frac{L_{\max}^2}{L_T^2}} - 1}{\sqrt{N}}\right).$$

It completes the proof. $\qquad\square$

**Proof to Corollary 3**

*Proof.* For ASGD in (11), letting $\omega = 0$ and $Y = N$ in $\alpha_1$, $\alpha_2$, and $\alpha_3$ in (7), we have

$$\alpha_1 = 4\left(1 + \frac{L_T^2(1+T)T}{L^2}\right),$$

$$\alpha_2 = \frac{1}{(f(x_0) - f^*)L},$$

$$\alpha_3 = \frac{L^2}{L_T^2}(K\sigma^2\alpha_2 + 4).$$

Next letting $\omega = 0$ and $Y = N$ in Theorem 1, the prerequisite for $T$ becomes

$$\begin{aligned}
T &\leqslant \frac{\sqrt{N}}{2\sqrt{N}}\left(\sqrt{1 + 4N^{-1/2}N^{1/2}\alpha_3} - 1\right) \\
&= \frac{1}{2}\left(\sqrt{1 + 4\alpha_3} - 1\right) \\
&= \frac{1}{2}\left(\sqrt{1 + 4\frac{L^2}{L_T^2}\left(\frac{K\sigma^2}{(f(x_0) - f^*)L} + 4\right)} - 1\right) \\
&= O\left(\sqrt{K\sigma^2 + 1}\right).
\end{aligned}$$

We finally obtain the following convergence rate

$$\frac{\sum_{k=0}^{K}\mathbb{E}\|\nabla f(x_k)\|^2}{2K}$$

$$\begin{aligned}
&\leqslant \frac{1}{K\alpha_2}\left(20 + \frac{L_T^2}{L_Y^2}\frac{\sqrt{Y}\left(\sqrt{1 + 4Y^{-1/2}N^{1/2}\alpha_3} - 1\right)}{\sqrt{N}}\right) \\
&\quad + \frac{11\sqrt{N\omega + \sigma^2}}{\sqrt{K\alpha_2}} + N\omega \\
&= \frac{1}{K\alpha_2}\left(20 + \frac{L_T^2}{L^2}\sqrt{1 + 4\alpha_3} - 1\right) + \frac{11\sigma}{\sqrt{K\alpha_2}} \\
&\leqslant \frac{1}{K\alpha_2}\left(20 + \frac{L_T^2}{L^2}\sqrt{5\alpha_3}\right) + \frac{11\sigma}{\sqrt{K\alpha_2}} \\
&\leqslant \frac{1}{K\alpha_2}\left(20 + \frac{L_T^2}{L^2}\sqrt{5\alpha_3}\right) + \frac{11\sigma}{\sqrt{K\alpha_2}} \\
&= \frac{1}{K\alpha_2}\left(20 + \frac{L_T^2}{L^2}\sqrt{5\frac{L^2}{L_T^2}(K\sigma^2\alpha_2 + 4)}\right) + \frac{11\sigma}{\sqrt{K\alpha_2}} \\
&= \frac{1}{K\alpha_2}\left(20 + \frac{L_T\sqrt{5}}{L}\sqrt{K\sigma^2\alpha_2 + 4}\right) + \frac{11\sigma}{\sqrt{K\alpha_2}} \\
&\leqslant \frac{1}{K\alpha_2}\left(20 + \frac{L_T\sqrt{5}}{L}\left(\sqrt{K\sigma^2\alpha_2} + 2\right)\right) + \frac{11\sigma}{\sqrt{K\alpha_2}} \\
&= O\left(\frac{1}{K} + \frac{\sigma}{\sqrt{K}}\right),
\end{aligned}$$

It completes the proof. $\qquad\square$

## Proof to Corollary 4 and Corollary 5

*Proof.* Corollary 4 can be considered a special case of Corollary 5 with $\omega = 0$. We only prove Corollary 5 here, which automatically implies Corollary 4. Letting $Y = 1$ in (7) and Theorem 1, we obtain

$$\gamma^{-1} = 2L_{\max}N\left(\sqrt{\frac{\alpha_1^2}{K(N\omega + \sigma^2)\alpha_2 + \alpha_1}} + \sqrt{K(N\omega + \sigma^2)\alpha_2}\right),$$

and

$$\alpha_1 = 4\left(1 + \frac{L_T^2\left(1 + \frac{T}{\sqrt{N}}\right)T}{L_{\max}^2 N}\right),$$

$$\alpha_2 = \frac{1}{(f(x_0) - f^*)L_{\max}N},$$

$$\alpha_3 = \frac{L_{\max}^2}{L_T^2}(K(N\omega + \sigma^2)\alpha_2 + 4),$$

$$\omega = \frac{\sum_{i=1}^N L_{(i)}^2\mu_i^2}{N}.$$

The prerequisite for $T$ in Theorem 1 becomes

$$
\begin{aligned}
T &\leqslant \frac{\sqrt{N}}{2}\left(\sqrt{1 + 4\alpha_3 N^{1/2}} - 1\right) \\
&= \frac{\sqrt{N}}{2}\left(\sqrt{1 + 4\frac{L_{\max}^2}{L_T^2}(K(N\omega + \sigma^2)\alpha_2 + 4)N^{1/2}} - 1\right) \\
&= O\left(N^{3/4}\sqrt{K\left(\omega + \frac{\sigma^2}{N}\right) + 1}\right) \\
&= O\left(\sqrt{N^{3/2} + KN^{1/2}\sigma^2}\right).
\end{aligned}
\tag{47}
$$

The convergence rate becomes

$$
\begin{aligned}
&\frac{\sum_{k=0}^K \mathbb{E}\|\nabla f(x_k)\|^2}{2K} \\
&\leqslant \frac{1}{K\alpha_2}\left(20 + \frac{L_T^2}{L_{\max}^2}\frac{\sqrt{1 + 4N^{1/2}\alpha_3} - 1}{\sqrt{N}}\right) + \frac{11\sqrt{N\omega + \sigma^2}}{\sqrt{K\alpha_2}} + N\omega \\
&= O\left(\frac{N}{K} + \frac{N^{3/4}}{\sqrt{K}}\sqrt{\omega + \frac{\sigma^2}{N} + \frac{1}{K}} + \frac{N\sqrt{\omega + \frac{\sigma^2}{N}}}{\sqrt{K}} + N\omega\right) \\
&\leqslant O\left(\frac{N}{K} + \frac{N^{3/4}}{\sqrt{K}}\left(\sqrt{\omega + \frac{\sigma^2}{N}} + \frac{1}{\sqrt{K}}\right) + \frac{N\sqrt{\omega + \frac{\sigma^2}{N}}}{\sqrt{K}} + N\omega\right) \\
&= O\left(\frac{N}{K} + \frac{\sqrt{N}\sqrt{N\omega + \sigma^2}}{\sqrt{K}} + N\omega\right) \\
&\leqslant O\left(\frac{N}{K} + \frac{\sqrt{N}\sigma}{\sqrt{K}} + \frac{N\sqrt{\omega}}{\sqrt{K}} + N\omega\right).
\end{aligned}
$$

Since $\omega = \frac{\sum_{i=1}^N L_{(i)}^2\mu_i^2}{N}$, if we use a constant $\mu$ for all $\mu_i$, and if we let $N\omega \leqslant O(N/K)$, $\frac{N\sqrt{\omega}}{\sqrt{K}} \leqslant O(N/K)$, we need

$$\mu \leqslant \sqrt{\frac{N}{K\sum_{i=1}^N L_{(i)}^2}} = O\left(\frac{1}{\sqrt{K}}\right).
\tag{48}$$

If let $N\omega \leqslant O(\frac{\sqrt{N}\sigma}{\sqrt{K}})$, $\frac{N\sqrt{\omega}}{\sqrt{K}} \leqslant O(\frac{\sqrt{N}\sigma}{\sqrt{K}})$, it suffices that

$$\mu \leqslant O\left(\min\left\{\frac{\sqrt{\sigma}}{(K)^{1/4}(N)^{1/4}}, \sigma/\sqrt{N}\right\}\right). \tag{49}$$

Since (18) satisfies either (48) or (49), we obtain a convergence rate of

$$\frac{\sum_{k=0}^{K}\mathbb{E}\|\nabla f(x_k)\|^2}{2K} \quad \leqslant \quad O\left(\frac{N}{K} + \frac{\sqrt{N}\sigma}{\sqrt{K}}\right).$$

The prerequisite (47) becomes

$$T \quad \leqslant \quad O\left(N^{3/4}\sqrt{K\left(\omega + \frac{\sigma^2}{N}\right) + 1}\right)$$

$$\leqslant \quad O\left(N^{3/4}\sqrt{1 + \frac{K\sigma^2}{N}}\right),$$

which completing the proof. $\qquad\square$