[Reviews · NeurIPS 2016]

Reviewer 1

Summary

The authors provide a comprehensive analysis of their zeroth-order method. They show that under special cases, their analysis improve the existing results. They have also done some experiments to verify their results.

Qualitative Assessment

The paper is clear overall. The literature review is comprehensive and serve as a good reference. It provides all the relevant results in a table. The paper also gave a generic analysis of asynchronous stochastic parallel optimization. It improved the existing results on special cases. However, the motivation for zeroth-order is not very clear. The paper emphasis that the zeroth-order algorithm is a novel contribution. Yet, it seems that the zeroth-order algorithm is in essence an approximation to the gradient descent. It might help if the authors could make it clear in what situation we only have access to the function but not the gradient. Also, a major part of the discussion (Corollary 2,3,4) is based on the case where w=0, but the zeroth-order algorithm should have a positive w in practice (w=0 is reduced to first order method). It would be beneficial if the authors can provide more insights on their zeroth-order algorithms. Comments: the vertical spacing is not right. Line 65, the definition of zeroth-order should be fined before.

Confidence in this Review

2-Confident (read it all; understood it all reasonably well)


Reviewer 2

Summary

The paper considers asynchronous parallelization of first- and zeroth-order stochastic descent algorithms, and analyzes the feasibility of their linear speedup. The general bound derived in Theorem 1 is then applied to both Stochastic Coordinate Descent and Stochastic Gradient Descent cases as corollaries, for which bounds on speedup are obtained that relate #workers, dimensionality, iteration index and stochastic gradient variance. The paper also obtains convergence rate for zeroth-order descent. Experiments evaluate the zeroth-order method in multi-core environment for a synthetic neural network, and for ensembling submodels on Yahoo! Music recommendation dataset.

Qualitative Assessment

The theoretical analysis in the paper is interesting due to the generalization it provides for a number of earlier methods, and for the new results for zeroth-order case. One blindspot is considering the effects of the infrastructure in the multi-node case: network bandwidth becomes highly contended, and corresponding effects should be accounted for in the analysis. The experimental validation in the paper can be improved significantly: - Most frequent practical application of ASCD/ASGD is for training large, highly sparse linear models for classification. It is unclear why this was not verified. - The use of the synthetic DNN is unwarranted given the wide popularity and accessibility of standard DNN benchmarks.

Confidence in this Review

2-Confident (read it all; understood it all reasonably well)


Reviewer 3

Summary

The paper provides a general analysis of a wide class of asynchronous stochastic algorithms, including stochastic gradient descent, and stochastic coordinate descent, all in the lock-free case. Serious numerical examples are provided to back up their claims of near linear speedup.

Qualitative Assessment

This is the best paper I have read for NIPS this year. I think it is a great contribution. Not only is a powerful framework and theorem provided, but the authors take the time to show how to apply it to other cases, and they discuss interesting features (for example, they make sure the results make sense in light of previous results, and discuss the effects of the variance in the gradient estimate). The authors instill confidence to the reader that the authors did a good job critically examining their own results. The numerical examples are not the typical small-scale test that claim this is the best algorithm ever, but rather they are well-down tests with good implementations on real-world datasets, and they do not claim to be the “best” algorithm ever, but rather test the stated claim of the paper — namely, that the asynchronous nature leads to speedup benefits up to a certain point. My main suggestion is that the authors re-proof the paper, as there are still many grammar mistakes (for example, many missing articles before nouns). I also must mention that I did not have time to read the appendix, so I am taking the proofs at the authors’ word. Other minor issues I noticed: - “sever” is listed several times, and should be “server” as in the title of [17]. - You might want to put the equation on line 107 as its own numbered equation, and mention that this is actually a change in the algorithm when mu=0. - Line 113, I don’t see where \hat{x}_k has been defined, so I am a bit confused about where this fits into the discussion. - Equation (4), I don’t see any coordinate indices, just the iteration indices, so it seems like this is a consistent read, not an inconsistent read. Maybe this is a notational issue? I’m confused. - Line 150, the choice of gamma for the Theorem to hold requires a lot of known parameters. Could you analyze the algorithm assuming that gamma is this quantity, up to a fixed constant? Or at least run numerical examples showing the robustness of the algorithm to incorrect choices in gamma? - Line 241 is not a complete sentence.

Confidence in this Review

2-Confident (read it all; understood it all reasonably well)


Reviewer 4

Summary

This paper provides a survey of existing asynchronous SGD algorithms. It then proposes a async zero order SGD and applies them to parameter selection and model blending problems. The paper performs staleness analysis on existing ASGD algorithms and shows that variance and model dimension affect speedup.

Qualitative Assessment

The paper is easy to read and follow. The authors provide an improved convergence analysis, in special cases. It is not clear to the reader why these additional analyses are useful. Your definition of inconsistent read assumes that the individual x values are consistent (Section 3.2). It is very much possible that when a float value is being written, another child reads it and it reads a garbage value for that float. You should state this as an assumption clearly. Likewise. reading and writing atomically can be guaranteed even with a multi-core implementation by using an atomic read/write library. Similarly, one can get inconsistent reads even within a parameter server if implemented using direct memory access protocols. Hence, statements like "However, in this [multicore]computational model atomic read and write of x cannot be guaranteed." needs to be replaced with "in popular implementations of this computational model atomic read and write of x is not guaranteed"

Confidence in this Review

1-Less confident (might not have understood significant parts)


Reviewer 5

Summary

This paper provides a generic analysis of asynchronous parallel stochastic algorithms including first order and zero order methods. Its contribution mainly focuses on the theoretical analysis of conditions ensuring the linear speedup property by T workers. The main theorem can cover results on existing algorithms such as ASCD and ASGD, and improves the analysis, in particular, T can be larger than that in previous work to guarantee the linear speedup. In addition, its generic analysis suggests a new zero order gradient descent algorithm (ASZD). The experiment result of ASZD on real dataset is given.

Qualitative Assessment

The paper is well written and pleasant to read. The contribution of this paper is two-fold. 1. It improves the analysis on the existing asynchronous parallel first order stochastic optimization. Its proof technique seems to have some novelty, although I did not check the detail of the proof and compare with existing ones. The author claims that ASCD and ASGD have been validated in several recent papers, but I still hope to look at some simple experiments on them and see whether it matches your tighter result or not, e.g., in corollary 4, it needs T\leq O(\sqrt{N^{3/2}+KN^{1/2}\sigma^2}) rather than O(\sqrt{KN^{1/2}\sigma^2}) in the previous work. 2. ASZD is proposed and tested on real dataset. The analysis is covered by the main theorem.

Confidence in this Review

2-Confident (read it all; understood it all reasonably well)


Reviewer 6

Summary

This paper unifies the analysis asynchronous SGD with asynchronous coordinate descent. It also proposes an asynchronous algorithm for derivative free optimization.

Qualitative Assessment

The behavior of asynchronous optimization algorithms is less well understood, and contributions to the theory in this case are valuable. The algorithm studied (Algorithm 1) is essentially a hybrid of SGD and coordinate descent, and so results about Algorithm 1 can be specialized to recover results about SGD and results about coordinate descent. However, it is not clear to me that this adds much value on top of the original analyses of SGD and coordinate descent or that Algorithm 1 is a particularly interesting algorithm to study. Does it really represent generic asynchronous optimization? Or is it mostly a unification of SGD and coordinate descent? For the zeroth order optimization case, the result is interesting. The authors demonstrate that this algorithm achieves a speedup over the single-core case, but is this a sensible approach relative to other zeroth-order optimization algorithms?

Confidence in this Review

1-Less confident (might not have understood significant parts)